# MASKED TEMPORAL INTERPOLATION DIFFUSION FOR PROCEDURE PLANNING IN INSTRUCTIONAL VIDEOS

**Yufan Zhou**[1]  **Zhaobo Qi**[1*]  **Lingshuai Lin**[1]  **Junqi Jing**[1]
**Tingting Chai**[1]  **Beichen Zhang**[1]  **Shuhui Wang**[2]  **Weigang Zhang**[1*]
[1]Harbin Institute of Technology, Weihai
[2]Key Lab of Intell. Info. Process., Inst. of Comput. Tech., CAS
{yfzhou, lslin, jqjing}@stu.hit.edu.cn, wangshuhui@ict.ac.cn,
{qizb, beiczhang, ttchai, wgzhang}@hit.edu.cn

## ABSTRACT

In this paper, we address the challenge of procedure planning in instructional videos, aiming to generate coherent and task-aligned action sequences from start and end visual observations. Previous work has mainly relied on text-level supervision to bridge the gap between observed states and unobserved actions, but it struggles with capturing intricate temporal relationships among actions. Building on these efforts, we propose the Masked Temporal Interpolation Diffusion (MTID) model that introduces a latent space temporal interpolation module within the diffusion model. This module leverages a learnable interpolation matrix to generate intermediate latent features, thereby augmenting visual supervision with richer mid-state details. By integrating this enriched supervision into the model, we enable end-to-end training tailored to task-specific requirements, significantly enhancing the model's capacity to predict temporally coherent action sequences. Additionally, we introduce an action-aware mask projection mechanism to restrict the action generation space, combined with a task-adaptive masked proximity loss to prioritize more accurate reasoning results close to the given start and end states over those in intermediate steps. Simultaneously, it filters out task-irrelevant action predictions, leading to contextually aware action sequences. Experimental results across three widely used benchmark datasets demonstrate that our MTID achieves promising action planning performance on most metrics. The code is available at https://github.com/WiserZhou/MTID.

## 1 INTRODUCTION

Recently, procedure planning has exhibited critical reasoning capability for solving real-world challenges in complex domains, such as robotic navigation (Sermanet et al., 2024; Bhaskara et al., 2024) and autonomous driving (Wang et al., 2024; Liao et al., 2024). Among them, procedure planning in instructional videos (Zhao et al., 2022; Wang et al., 2023b; Li et al., 2023) has been widely concerned because of its wide application scenarios, which involve identifying and generating coherent action sequences that align with the task's objectives, given the start and end visual observations.

In the field of procedure planning in instructional videos, the primary challenge lies in modeling the temporal evolution mechanism among actions and identifying pertinent conditions that can effectively steer the generation of intermediary actions in scenarios where information is scarce. As depicted in Figure 1(a), many scholars have resorted to capturing different forms of auxiliary information about the intermediate states to bridge the gap between observed states and unobserved actions. For example, event-based supervision (Wang et al., 2023a) leverages key task events to help the model learn temporal action structures, while task label supervision (Wang et al., 2023b) uses task-specific labels for better alignment with the task objective. The probabilistic procedure knowledge graph (Nagasinghe et al., 2024) provides structured knowledge to enhance the understanding of action dependencies. Additionally, Niu et al. (2024) leverage large language models (LLMs) to describe state changes, improving the model's grasp of causal relationships by combining visual

---

*Corresponding Authors.

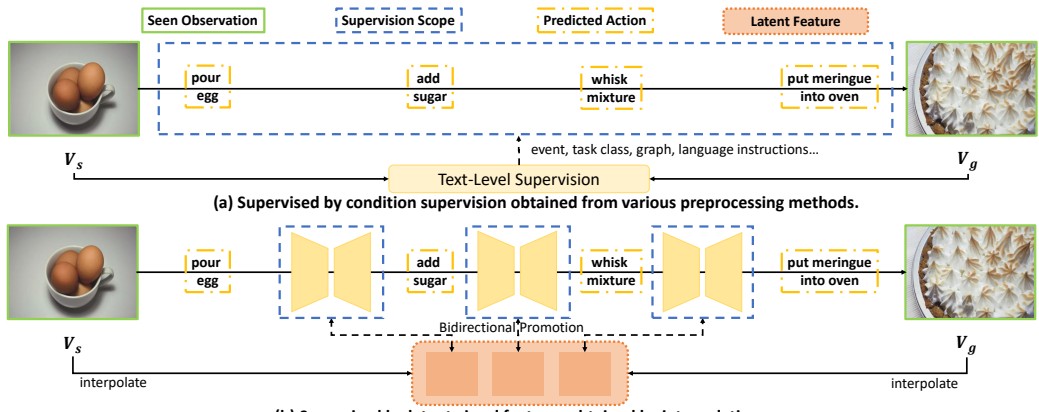

Figure 1: The core idea to solve procedure planning with previous methods and ours.

and language descriptions. However, all these methods are limited to providing text-level supervision, resulting in less detailed and comprehensive information, and failing to precisely capture the temporal relationships between actions. Furthermore, these methods decouple the acquisition of supervisory information from the intermediate action reasoning process, hindering effective collaboration and interaction between the two. Consequently, it becomes challenging to fully integrate and adapt to current action reasoning tasks.

Based on the above analysis, we propose the Masked Temporal Interpolation Diffusion (MTID) model for procedure planning in instructional videos. As shown in Figure 1(b), the core concept is to leverage intermediate latent visual features, generated synchronously by a latent space temporal interpolation module, to provide comprehensive visual-level information for mid-state supervision. In the meanwhile, the generated visual features are directly injected into the action reasoning model, ensuring the generation of intermediate supervision information can be effectively applied to the current action reasoning task through end-to-end training.

Specifically, **MTID** comprises three core components: a task classifier, a latent space temporal interpolation module, and a diffusion model framework that integrates the DDIM strategy. In the first stage, a transformer-based classifier predicts the task class label $c$ for the entire instructional video, given the start and end observations. This prediction serves as the foundation for subsequent action generation. The latent space temporal interpolation module is designed to capture and model temporal relationships. It employs an observation encoder to transform the visual features of the observations into latent features that maintain temporal dependencies. A latent space interpolator then generates intermediate features using a learnable interpolation matrix, which dynamically adjusts the interpolation ratio to fit task-specific requirements. These interpolated features are refined through transformer blocks, enhancing their temporal coherence and capturing the dependencies between action sequences. In the third stage, during the denoising phase for generating action sequences, the input matrix is constructed by concatenating the task class label, observed visual features, and Gaussian noise. A masked projection is applied to exclude irrelevant actions, ensuring that the generated actions remain within the desired range. To accelerate inference, DDIM is used throughout the iterative process. To further ensure task relevance, we adopt a task-adaptive masked proximity loss which gradually decreases its focus toward the central features, reinforcing supervision on intermediate latent features while penalizing irrelevant actions, thereby constraining the generation process. By leveraging both the start and end observations, our model accurately predicts target action sequences, as demonstrated by experimental results on the CrossTask, COIN, and NIV datasets.

The main contributions of this paper are as follows:

- We propose a Masked Temporal Interpolation Diffusion model with a mask to limit action initialization and a task-adaptive masked proximity loss to enhance accuracy.

- We use a latent space temporal interpolation module to extract intermediate visual features with temporal relationships from the start and end states to guide the diffusion process.

- Extensive experiments are conducted on several widely used benchmarks, showing significant performance improvements on multiple tasks using the proposed method.

## 2 RELATED WORK

**Procedure Planning in Instructional Videos.** Procedure planning involves generating goal-directed action sequences from visual observations in unstructured videos, like Qi et al. (2021; 2024). Our work builds on PDPP (Wang et al., 2023b), which models action sequences using diffusion processes. Earlier approaches focus on learning sequential latent spaces (Chang et al., 2020) and adversarial policy learning (Bi et al., 2021). Recent methods introduce linguistic supervision for step prediction (Zhao et al., 2022), mask-and-predict strategies for step relationships (Wang et al., 2023a), and breaking sequences into sub-chains by skipping unreliable actions (Li et al., 2023). KEPP (Nagasinghe et al., 2024) incorporates probabilistic knowledge for step sequencing, while SCHEMA (Niu et al., 2024) tracks state changes at each step. However, none of these methods focus on the visual-level temporal logic between actions. Our approach introduces mid-state temporal supervision to capture these relationships, resulting in more accurate and efficient predictions.

**Diffusion Probabilistic Models for Long Video Generation.** Recent advances in diffusion probabilistic models (Croitoru et al., 2023), originally popularized for image generation (Rombach et al., 2022), have achieved significant progress in generating long video sequences (Weng et al., 2024; Zhou et al., 2024a; Jiang et al., 2024). StreamingT2V (Henschel et al., 2024) excels in producing temporally consistent long videos with smooth transitions and high frame quality, overcoming the typical limitations of short video generation. StoryDiffusion (Zhou et al., 2024b) further enhances sequence coherence through consistent self-attention, enabling the creation of detailed, visually coherent stories. The success of these models has inspired us to incorporate auxiliary temporal coherence mechanisms in our approach, which we believe are critical to achieving accurate and high-quality action prediction with our method.

## 3 METHOD

### 3.1 OVERVIEW

Following Chang et al. (2020), given an initial visual observation $V_s$ and a target visual observation $V_g$, both are short video clips indicating different states of the real-world environment extracted from an instructional video, the procedure planning task aims to generate a sequence of actions $a_{1:T}$ that transforms the environment from $V_s$ to $V_g$, where $T$ denotes the number of planning time steps. This problem can be formulated as $p(a_{1:T} \mid V_s, V_g)$.

Considering the weak temporal reasoning ability caused by the absence of intermediate visual states, especially in long video scenarios, we propose the **Masked Temporal Interpolation Diffusion** (MTID) framework, which employs a denoising diffusion model to rapidly predict the intermediate action sequence $a_{1:T}$. As outlined in the following formula, MTID decomposes the procedure planning task into three sub-problems,

$$p(a_{1:T} \mid V_s, V_g) = \iint p(a_{1:T} \mid v_{1:M}, c, V_s, V_g) p(v_{1:M} \mid V_s, V_g) p(c \mid V_s, V_g) dv_{1:M} dc. \quad (1)$$

The first sub-problem entails capturing information about the task to be completed about the whole instructional video, serving as the basis for subsequent reasoning. As shown in Figure 2, this task supervision stage solves a standard classification problem using a transformer encoder to extract features from observation pair $\{V_s, V_g\}$ and transform them into task class label $c$.

The second sub-problem focuses on reconstructing $M$ intermediate visual features $v_{1:M}$ from $V_s$ and $V_g$ to address the lack of mid-state visual supervision and reveal hidden temporal logic within the action sequences, which is achieved through our latent space temporal interpolation module.

The final sub-problem involves generating action sequence $a_{1:T}$ based on the task information and interpolated intermediate features. Specifically, we first construct the input matrix $\hat{x}_N$ for the denoising steps, which consists of three dimensions. The task class dimension contains the captured task information $c$ for each reasoning step. The observation dimension contains the visual observations of the start and goal states $\{V_s, V_g\}$, where the intermediate states are set to zero. The action dimension contains $\hat{a}_{1:T}$ which represents the intermediate state action sequence, and is initialized by $\epsilon \sim \mathcal{N}(0, I)$ and further constrained by our action mask mechanism to reduce the action space

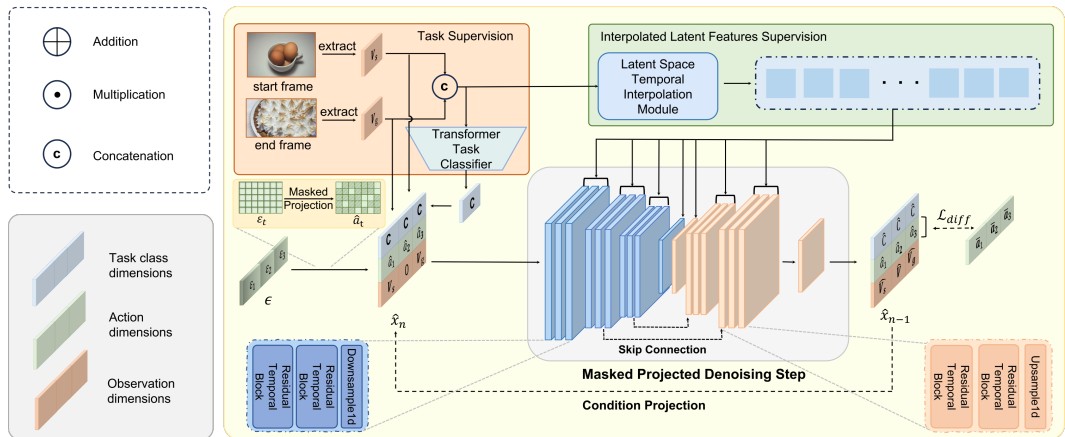

Figure 2: Overview of our Masked Temporal Interpolation Diffusion (prediction horizon $T = 3$). With a transformer classifier to guide from observations, $\hat{x}_n$ is processed in a U-Net with latent space module for temporal supervision. Starting with denoising through masked projection, we derive the final actions needed to compute the task-adaptive masked proximity loss for iterative optimization.

to be predicted. Hence, the iteration matrix $\hat{x}_n \in \mathbb{R}^{T \times (C+A+O)}$ is expressed as,

$$\hat{x}_n = \begin{bmatrix} c & c & \cdots & c & c \\ \hat{a}_1 & \hat{a}_2 & \cdots & \hat{a}_{T-1} & \hat{a}_T \\ V_s & 0 & \cdots & 0 & V_g \end{bmatrix}, \tag{2}$$

where $n$ ranges from 0 to $N$, $C$ is the number of task classes, $A$ is the number of actions and $O$ is the observation visual feature dimension. Next, the intermediate latent features generated from the latent space temporal interpolation module will be injected into the diffusion model to iteratively optimize the matrix $\hat{x}_n$. During the iteration process, we adopt DDIM to accelerate the sampling process with fewer steps while maintaining strong performance. Lastly, we introduce the task-adaptive masked proximity loss $\mathcal{L}_{\text{diff}}$ to enhance the reliability of the reasoning results.

## 3.2 MTID: MASKED TEMPORAL INTERPOLATION DIFFUSION

### 3.2.1 PRELIMINARIES

Denoising Diffusion Implicit Model (DDIM) (Song et al., 2021) improves sampling efficiency by making the reverse process deterministic, which reduces stochastic noise and establishes a direct mapping between the initial noise matrix $\hat{x}_N$ and the final output matrix $\hat{x}_0$ across $N$ non-Markovian steps. This approach reduces the number of steps needed while preserving sample quality.

Based on these advantages, we adopt the DDIM sampling strategy with the U-Net model (Ronneberger et al., 2015) for its ability to accelerate sampling with fewer steps while maintaining strong performance. This is especially useful in scenarios where the quality of results remains comparable to that of Denoising Diffusion Probabilistic Model (DDPM) (Ho et al., 2020; Nichol & Dhariwal, 2021), despite its deterministic approach.

The forward process is parameterized as:

$$\hat{x}_N = \sqrt{\bar{\alpha}_N}\, \hat{x}_0 + \sqrt{1 - \bar{\alpha}_N}\, \epsilon, \tag{3}$$

where $\bar{\alpha}_N = \prod_{s=1}^{N} \alpha_s$, $\epsilon \sim \mathcal{N}(0, I)$, and $\alpha_s = 1 - \beta_s$, which represents the noise schedule controlling the amount of Gaussian noise added at each step $s$. The forward process starts with the original data $\hat{x}_0$ and progressively adds noise, resulting in the final noisy matrix $\hat{x}_N$.

In DDIM, the reverse process is defined as:

$$\hat{x}_{n-1} = \sqrt{\bar{\alpha}_{n-1}} \left( \frac{\hat{x}_n - \sqrt{1 - \bar{\alpha}_n} f_\theta\left(\hat{x}_n\right)}{\sqrt{\bar{\alpha}_n}} \right) + \sqrt{1 - \bar{\alpha}_{n-1}} \cdot f_\theta\left(\hat{x}_n\right), \tag{4}$$

where $f_\theta\left(\hat{x}_n\right)$ is the neural network's prediction of the noise component added to $\hat{x}_n$. This reverse process reconstructs the original data $\hat{x}_0$ from $\hat{x}_N$ by iteratively removing the noise introduced during the forward diffusion process. Unlike DDPM, DDIM's deterministic reverse process improves

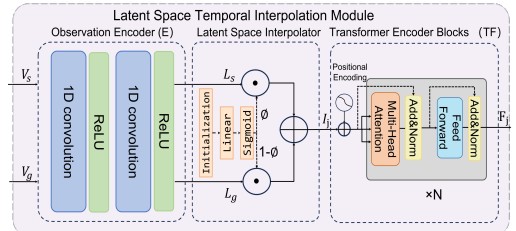 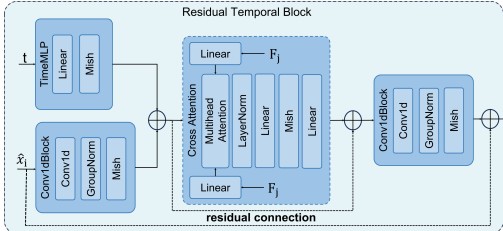

(a) Latent space temporal interpolation module.  (b) Residual temporal block & cross-attention module.

Figure 3: Module in Figure 3a generates temporally coherent latent features from observations, while the block in Figure 3b refines these features with temporal dependencies, guiding coherent action sequence generation.

sampling efficiency by directly mapping the noisy input $\hat{x}_N$ to the final output $\hat{x}_0$. This makes DDIM an efficient method for generating or enhancing samples with fewer steps.

### 3.2.2 LATENT SPACE TEMPORAL INTERPOLATION

As shown in Figure 3a, the **Latent Space Temporal Interpolation Module** consists of three core components: an observation encoder, which transforms observation visual features into latent features; a latent space interpolator, which generates multiple features to fill the intermediate supervision; and transformer encoder blocks, which refine the generated features and enhance the module's ability to model the temporal correlations among action sequences.

Specifically, this module first employs an observation encoder $E$, consisting of convolutional layers, to transform the visual observations $V_s$ and $V_g$ into their respective latent features $L_s$ and $L_g$:

$$L_s, L_g = E(V_s, V_g). \tag{5}$$

In the latent space, we perform linear interpolation between $L_s$ and $L_g$ to generate a sequence of interpolated features $\{I_1, I_2, \ldots, I_M\}$. Unlike fixed linear interpolation, our method dynamically adjusts the interpolation through the learnable interpolation matrix $\phi \in \mathbb{R}^{M \times O}$ to facilitate smooth and task-specific transitions. This matrix, restricted between 0 and 1, is responsible for weighting the latent features $L_s$ and $L_g$ to generate the intermediate latent features as follows:

$$\begin{aligned} \phi &= Sigmoid(W \cdot \tau + k), \\ I_j &= (1 - \phi_j) \cdot L_s + \phi_j \cdot L_g, \end{aligned} \tag{6}$$

where $W \in \mathbb{R}^{M \times O}$ and $k \in \mathbb{R}^{M \times O}$ are the parameters of the linear layer, and $O$ represents the observation dimension. The matrix $\tau \in \mathbb{R}^{M \times O}$ is a learnable matrix initialized with a constant value, controlling a variable ratio between $L_s$ and $L_g$. This method requires no parameter tuning and offers better adaptability to different tasks. The number of latent features $M$ depends on the number of residual temporal blocks in the model.

The sequence of interpolated features $\{I_1, I_2, \ldots, I_M\}$ is then passed through a series of transformer encoder blocks $TF$ to obtain the enhanced latent features:

$$F_1, F_2, \ldots, F_M = TF(I_1, I_2, \ldots, I_M). \tag{7}$$

The self-attention mechanism in the transformers captures dependencies between latent features at different time steps by computing attention scores across all latent features. Stacking multiple transformer blocks allows the model to iteratively refine these features, ensuring that temporal and contextual relationships are effectively learned.

To integrate the interpolated latent features $\{F_1, F_2, \ldots, F_M\}$ into the model during the denoising process, we incorporate cross-attention layers (Khachatryan et al., 2023) into the residual temporal blocks of the U-Net, as shown in Figure 3b. This allows the model to dynamically focus on relevant latent features through a learnable matrix, enhancing its ability to capture complex relationships with more latent temporal information and improving the quality of action predictions.

In this setup, the latent feature $F_j$, processed through a linear layer, serves as the key and value, while $\hat{x}_n$, processed through a convolutional block and combined with $t$ (sampled from random

integers), acts as the query. The cross-attention is computed as:

$$\hat{x}_n = softmax \left( \frac{[CB(\hat{x}_n) + TM(t)] \cdot LF_j^T}{\sqrt{O}} \right) \cdot LF_j, \tag{8}$$

where $CB$ denotes a 1D convolution block, $TM$ represents a time MLP, and $LF_j$ refers to the result obtained from passing $F_j$ through a linear layer. This design enables the model to effectively capture temporal relationships, thereby improving the overall quality of action prediction.

### 3.2.3 MASKED PROJECTION FOR INITIALIZATION

While our model handles procedure planning tasks effectively, the denoising sampling process in diffusion models does not always guarantee that the generated actions fall within the desired range. To mitigate this issue, we introduce a masked projection that constrains the action space during the training denoising process. This approach is inspired by the masked latent modeling scheme proposed by Gao et al. (2023).

Since each task has a specific action scope, we activate only the actions associated with the current task class label $c$ and deactivate the others. When constructing the input matrix $\hat{x}_N$ for the denoising process, we add the initial Gaussian noise solely to the active action positions, while setting the non-active action positions to zero. This process can be expressed as follows:

$$\hat{a}_{t,d} = \begin{cases} \epsilon, & \text{if } d \in Task(c) \\ 0, & \text{if } d \notin Task(c) \end{cases}, \tag{9}$$

where $d$ denotes the action ID spanning the action dimension $A$, and $\epsilon \sim \mathcal{N}(0, 1)$. The function $Task(c)$ represents the set of actions corresponding to task $c$. By restricting the initial noise in the input matrix $\hat{x}_N$ to the relevant action scope, the model ensures that the procedure planning is confined to the active actions.

### 3.3 TASK-ADAPTIVE MASKED PROXIMITY LOSS

Our training process consists of two main stages: (a) training a task classifier to extract task-related information based on the given start and goal visual observations. (b) utilizing a masked temporal interpolation diffusion model $f_\theta$ to fit the distribution of the target action sequence.

In the first stage, we minimize the cross-entropy loss between the predicted and true task classes to optimize the transformer-based task classifier.

In the second stage, we employ a diffusion-based training scheme and introduce a task-adaptive masked proximity loss to model the target action sequence, defined as follows:

$$\mathcal{L}_{\text{diff}} = \sum_{t=1}^{T} \sum_{d=1}^{A} w_t \cdot m_{t,d} \cdot (a_{t,d} - \bar{a}_{t,d})^2, \tag{10}$$

where $a_{t,d}$ refers to the predicted action ID extracted from the final output $\hat{x}_0$, and $\bar{a}_{t,d}$ denotes the ground truth action. This loss function computes the weighted mean squared error (MSE) between the predicted and ground truth actions at each planning time step. The term $w_t$ is a time-dependent gradient weight that controls the contribution of each time step, and $m_{t,d}$ is a mask matrix that highlights specific action dimensions or planning time steps according to the task requirements.

The weight $w_t$ of gradient weighted loss is defined as:

$$w_t = w_0 + (1 - w_0) \cdot \frac{\min(t, T - t + 1) - 1}{\lceil T/2 \rceil - 1}, \tag{11}$$

where $w_0$ is the initial weight. Since the task only observes the start and goal features, $V_s$ and $V_g$, higher weights are assigned to predictions near these endpoints, thereby enhancing performance at $a_1$ and $a_T$. Lower weights are assigned to the intermediate steps, allowing the model to balance the endpoints and middle states without placing too much emphasis on the endpoints. Unlike Wang et al. (2023b), who weights both start and end actions called both sides weighted loss, our approach uses intermediate latent features for continuous supervision. This provides more comprehensive guidance, allowing us to apply gradient weights for better alignment of the entire action sequence.

Table 1: Comparison with other methods on CrossTask dataset. Features extracted by the HowTo100M-trained encoder and settings of PDPP are marked with †, while other features are provided directly by CrossTask. **Note that we compute mIoU by calculating average of every IoU of a single action sequence rather than a mini-batch for all datasets.**

| Models | T = 3 | | | T = 4 | | | T=5 | T = 6 |
|---|---|---|---|---|---|---|---|---|
| | SR↑ | mAcc↑ | mIoU↑ | SR↑ | mAcc↑ | mIoU↑ | SR↑ | SR↑ |
| Random | <0.01 | 0.94 | 1.66 | <0.01 | 0.83 | 1.66 | <0.01 | <0.01 |
| Retrieval-Based | 8.05 | 23.30 | 32.06 | 3.95 | 22.22 | 36.97 | 2.40 | 1.10 |
| WLTDO (Ehsani et al., 2018) | 1.87 | 21.64 | 31.70 | 0.77 | 17.92 | 26.43 | — | — |
| UAAA (Abu Farha & Gall, 2019) | 2.15 | 20.21 | 30.87 | 0.98 | 19.86 | 27.09 | — | — |
| UPN (Srinivas et al., 2018) | 2.89 | 24.39 | 31.56 | 1.19 | 21.59 | 27.85 | — | — |
| DDN (Chang et al., 2020) | 12.18 | 31.29 | 47.48 | 5.97 | 27.10 | 48.46 | 3.10 | 1.20 |
| PlaTe (Sun et al., 2022) | 16.00 | 36.17 | 65.91 | 14.00 | 35.29 | 55.36 | — | — |
| Ext-GAIL (Bi et al., 2021) | 21.27 | 49.46 | 61.70 | 16.41 | 43.05 | 60.93 | — | — |
| P³IV (Zhao et al., 2022) | 23.34 | 49.96 | 73.89 | 13.40 | 44.16 | 70.01 | 7.21 | 4.40 |
| EGPP (Wang et al., 2023a) | 26.40 | 53.02 | 74.05 | 16.49 | 48.00 | 70.16 | 8.96 | 5.76 |
| PDPP† (Wang et al., 2023b) | 37.2 | 64.67 | 66.57 | 21.48 | 57.82 | 65.13 | 13.45 | 8.41 |
| KEPP† (Nagasinghe et al., 2024) | 38.12 | 64.74 | 67.15 | 24.15 | 59.05 | 66.64 | 14.20 | 9.27 |
| SCHEMA† (Niu et al., 2024) | 38.93 | 63.80 | **79.82** | 24.50 | 58.48 | **76.48** | 14.75 | **10.53** |
| MTID (Ours)† | **40.45** | **67.19** | **69.17** | **24.76** | **60.69** | **67.67** | **15.26** | 10.30 |

Additionally, a mask matrix $m_{t,d}$ is applied to selectively emphasize certain planning time steps and action dimensions. This matrix is defined as:

$$m_{t,d} = \begin{cases} \rho, & \text{if } \hat{a}_{t,d} \text{ is active} \\ 1, & \text{otherwise} \end{cases}, \tag{12}$$

where $\rho$ is a scaling coefficient applied when the action is active, thereby increasing the penalty for unrelated actions. By this mechanism, actions that are unrelated to the current task are discouraged from appearing in the output, ultimately enhancing planning accuracy.

# 4 EXPERIMENTS

## 4.1 EVALUATION PROTOCOL

**Datasets and Settings.** We evaluate our MTID method on three instructional video datasets: **CrossTask** (Zhukov et al., 2019), **COIN** (Tang et al., 2019), and **NIV** (Alayrac et al., 2016). CrossTask consists of 2,750 videos across 18 tasks, covering 105 actions, with an average of 7.6 actions per video. COIN contains 11,827 videos spanning 180 tasks, with an average of 3.6 actions per video. NIV includes 150 videos from 5 tasks, with an average of 9.5 actions per video. We randomly split each dataset into training (70% of videos per task) and testing (30%), following previous works (Sun et al., 2022; Wang et al., 2023b; Niu et al., 2024). Furthermore, we conduct all experiments using the setting of PDPP, except for Table 3, where we adopt the setting of KEPP to ensure a fair comparison. For details, please refer to Appendix D for the comparison with PDPP.

**Metrics.** Following previous works (Sun et al., 2022; Zhao et al., 2022; Wang et al., 2023b; Niu et al., 2024; Nagasinghe et al., 2024), we evaluate the models using three key metrics: (1) **Success Rate (SR)** is the strictest metric, where a procedure is considered successful only if every predicted action step exactly matches the ground truth. (2) **mean Accuracy (mAcc)** computes the average accuracy of predicted actions at each time step, where an action is deemed correct if it matches the ground truth action at the corresponding time step. (3) **mean Intersection over Union (mIoU)** quantifies the overlap between the predicted procedure and the ground truth by calculating mIoU as $\frac{|\{a_t\} \cap \{\hat{a}_t\}|}{|\{a_t\} \cup \{\hat{a}_t\}|}$, where $\{a_t\}$ represents the set of ground truth actions, and $\{\hat{a}_t\}$ denotes the set of predicted actions.

## 4.2 RESULTS

**Results for Task Classifier.** The first stage of our approach involves predicting the task class based on the given start and goal observations. We implement this using transformer models, replacing

Table 2: Classification results on all datasets.

| Models | CrossTask$_{How}$ | | | | COIN | | NIV | |
| | $T=3$ | $T=4$ | $T=5$ | $T=6$ | $T=3$ | $T=4$ | $T=3$ | $T=4$ |
|---|---|---|---|---|---|---|---|---|
| PDPP | 92.43 | 92.98 | 93.39 | 93.20 | 79.42 | 79.42 | **100.00** | **100.00** |
| MTID (Ours) | **93.67** | **94.03** | **94.02** | **94.26** | **81.47** | **81.47** | **100.00** | **100.00** |

Table 3: Comparisons on COIN and NIV datasets. Note: only this table uses the KEPP's settings.

| Models | COIN | | | | | | NIV | | | | | |
| | $T=3$ | | | $T=4$ | | | $T=3$ | | | $T=4$ | | |
| | SR↑ | mAcc↑ | mIoU↑ | SR↑ | mAcc↑ | mIoU↑ | SR↑ | mAcc↑ | mIoU↑ | SR↑ | mAcc↑ | mIoU↑ |
|---|---|---|---|---|---|---|---|---|---|---|---|---|
| Random | <0.01 | <0.01 | 2.47 | <0.01 | <0.01 | 2.32 | 2.21 | 4.07 | 6.09 | 1.12 | 2.73 | 5.84 |
| DDN | 13.90 | 20.19 | 64.78 | 11.13 | 17.71 | 68.06 | 18.41 | 32.54 | 56.56 | 15.97 | 27.09 | 53.84 |
| P$^3$IV | 15.40 | 21.67 | 76.31 | 11.32 | 18.85 | 70.53 | 24.68 | 49.01 | 74.29 | 20.14 | 38.36 | 67.29 |
| EGPP | 19.57 | 31.42 | 84.95 | 13.59 | 26.72 | 84.72 | 26.05 | **51.24** | 75.81 | 21.37 | 41.96 | 74.90 |
| PDPP | 19.42 | 43.44 | - | 13.67 | 42.58 | - | 22.22 | 39.50 | 86.66 | 21.30 | 39.24 | 84.96 |
| KEPP | 20.25 | 39.87 | 51.72 | 15.63 | 39.53 | 53.27 | 24.44 | 43.46 | 86.67 | 22.71 | 41.59 | 91.49 |
| SCHEMA | **32.09** | 49.84 | 83.83 | 22.02 | 45.33 | 83.47 | 27.93 | 41.64 | 76.77 | 23.26 | 39.93 | 76.75 |
| MTID | 30.44 | **51.70** | **59.74** | **22.74** | **49.90** | **61.25** | **28.52** | 44.44 | **56.46** | **24.89** | **44.54** | **57.46** |

the two-layer Res-MLP architecture employed in Wang et al. (2023b), and train it using a simple cross-entropy (CE) loss. The classification results are presented in Table 2. Our method consistently outperforms previous approaches in all evaluated aspects.

**Comparisons on CrossTask.** We evaluate performance on CrossTask using four prediction horizons. The results in Table 1 demonstrate that our method outperforms all other approaches across all metrics, except for the SR at $T=6$, where our model ranks second. These improvements are consistent across longer prediction horizons ($T=4,5,6$) and other step-level metrics, including mAcc and mIoU.

**Comparisons on NIV and COIN.** Table 3 presents our evaluation results on the NIV and COIN datasets, demonstrating that our approach consistently outperforms or matches the best-performing methods across both datasets. These results highlight that our model performs robustly across datasets of varying sizes.

## 4.3 ABLATION STUDIES

**Task-Adaptive Masked Proximity Loss.** Table 4 demonstrates the effectiveness of our proposed loss strategy with $T=5$ on the CrossTask dataset, using Mean Squared Error (MSE) as the base loss. The results show that both the task-adaptive mask and gradient weighted loss improve performance. While MSE alone results in lower scores, adding masks and fixed weights provides moderate improvement. Our approach, which incorporates gradient weighted loss and intermediate supervision, significantly boosts performance by leveraging richer task-relevant features.

Table 4: Ablation studies on our loss function. Note: W: Both Sides Weighted Loss, GW: Gradient Weighted Loss, M: Mask.

| ID | MSE | W | GW | M | SR↑ |
|---|---|---|---|---|---|
| 1 | ✓ | | | | 11.89 |
| 2 | ✓ | ✓ | | | 13.90 |
| 3 | ✓ | | ✓ | | 15.10 |
| 4 | ✓ | | | ✓ | 13.26 |
| 5 | ✓ | ✓ | | ✓ | 13.93 |
| 6 | ✓ | | ✓ | ✓ | **15.26** |

**Masked Projection.** Table 5 demonstrates that our masked projection (MP) on $\hat{x}_N$ as input to the U-Net enhances performance by filtering out irrelevant actions, allowing the model to focus on task-relevant actions. We also experiment with applying the mask during denoising iterations, but this approach proved ineffective. During the denoising process, the input matrix contains both positive and negative logits, and

Table 5: Ablation study on projection and phase when $T=3$ on CrossTask dataset. Note: "CP" denotes the condition projection.

| Models | SR↑ | mAcc↑ | mIoU↑ |
|---|---|---|---|
| w/o MP | 39.17 | 66.49 | 68.38 |
| MP on iteration | 3.38 | 10.17 | 9.66 |
| MP on initialization | **40.45** | **67.19** | **69.17** |

in some cases, negative values can improve the final score. Masking at this stage disrupted the natural behavior of the logits and the diffusion denoising process. Furthermore, applying the mask

before the condition projection led to sub-optimal results due to inaccurate task labels. Therefore, we apply the mask only after the condition projection for optimal performance.

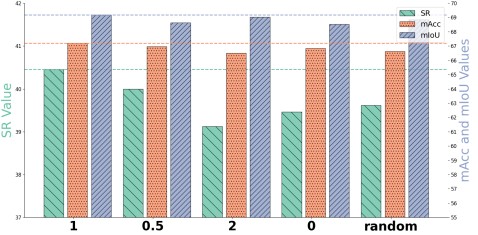

(a) Ablation studies on different simple initialization coefficient values $\tau$.

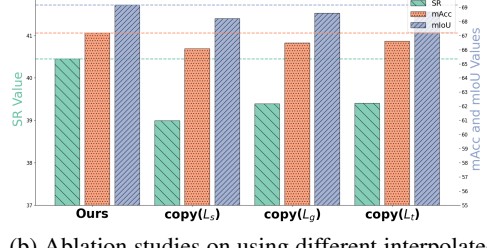

(b) Ablation studies on using different interpolated features for transformer blocks.

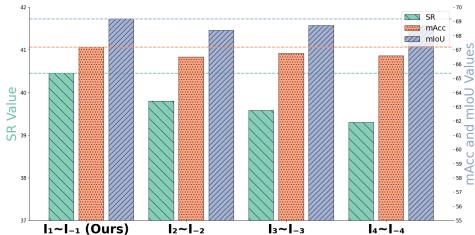

(c) Ablation studies on selection with different interpolated features.

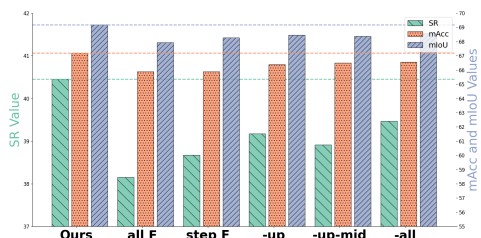

(d) Ablation studies on different fusion methods.

Figure 4: Ablation Studies for Interpolation Strategy. Figure 4b illustrates the features generated by the interpolator, where "$copy(L_g)$" means $I_j = L_g$, and "$copy(L_t)$" indicates $I_j = L_s$ for $j \leq \frac{M}{2}$ and $I_j = L_g$ for $j > \frac{M}{2}$. In Figure 4c, $I_i$ to $I_{-i}$ indicates that we use interpolated features from the $i$-th to the last $i$-th for the second interpolation. In Figure 4d, "all F" denotes that each cross-attention module receives $F_{1:M}$ as input, "step F" means we input one $F_{\frac{M}{2}}$ at the start and gradually increase the number of features inputted towards both sides, and "-all" indicates that we omit inputting $F_j$ during the upsampling, middle-sampling, and downsampling processes in U-Net.

**Latent Space Temporal Interpolation Module.** To evaluate the impact of various components within the module, we conduct ablation studies on the CrossTask dataset with $T = 3$. The results in Table 6 indicate that the observation encoder effectively transforms visual observations into latent features, enhancing causal inference and capturing temporal logic. The

Table 6: Component ablation. Note: Int: Interpolation, Enc: Encoder, Trans: Transformer.

| ID | Int | Enc | Trans | SR↑ | mAcc↑ | mIoU↑ |
|----|-----|-----|-------|------|-------|-------|
| 1 | ✓ | | | 37.86 | 65.42 | 67.32 |
| 2 | ✓ | ✓ | | 39.23 | 66.62 | 68.44 |
| 3 | ✓ | | ✓ | 39.49 | 66.81 | 68.68 |
| 4 | ✓ | ✓ | ✓ | **40.45** | **67.19** | **69.17** |

interpolator enriches these features by generating multiple interpolated versions, which aids in reasoning. The transformer encoder further refines these features, ensuring both mathematical interpolation and logical consistency, thereby improving the model's inference capabilities.

**Interpolation Strategy.** Figure 4 illustrates our adapted interpolation strategies for contrast. Initially, we test different initialization $\tau$ values in Figure 4a. The highest score occurs when $\phi$ is initialized to 1, which proves to be the most stable and achieves the best overall performance, indicating that $\phi$ converges close to 1. Further experiments in Figure 4b compare the use of $L_s$, $L_g$, or a combination of both through copying and direct return. The results show that directly returning $L_g$ performs well, suggesting that $V_g$ may play a critical role in action sequence inference. Although $L_s$ and $L_g$ are unadjusted features, the transformer encoder blocks can refine them to capture richer temporal logic and filter out irrelevant details, resulting in relatively reasonable scores. Additionally, we perform a second interpolation using the obtained $F_j$ (Figure 4c). This experiment shows that as the features shift towards the center (from $F_i$ to $F_{-i}$), performance declines due to the loss of original information, aligning with intuition. In Figure 4d, we examine different strategies for cross-attention. The "all F" approach results in sub-optimal performance, potentially due to the inclusion

of numerous features being limited by the capacity of the layers, which diminishes the amount of information each feature can carry and introduces disorder. Similarly, the "step F" strategy may suffer from the same issue. When cross-attention is removed from various stages of the U-Net, we observe that the more layers are removed, the worse the results become. However, when all layers are removed, the results improve slightly, suggesting that introducing latent features into the U-Net without sufficient information disturbs the original distribution, leading to sub-optimal outcomes.

## 4.4 UNCERTAINTY MODELING

We present the uncertainty modeling results for CrossTask and NIV in Table 7. Two baselines from Wang et al. (2023b) are used for comparison: the Noise baseline, which samples from random noise and uses the given observations and task class condition to obtain results in a single step without the diffusion process, and the Deterministic baseline, where $\hat{x}_N = 0$ and the model predicts a fixed outcome. We evaluate performance using KL divergence, NLL, ModeRec, and ModePrec, as outlined in Zhao et al. (2022).

Our approach outperforms the baselines on CrossTask, particularly in modeling uncertainty and generating diverse plans. Compared to Wang et al. (2023b), our model excels in the deterministic setting with $T = 3$, demonstrating its ability to capture latent temporal relationships even with fewer time steps. For the NIV dataset, we observe that despite its small size, our diffusion-based process still delivers improvements. Additional visualizations are provided in the appendix.

Table 7: The results of uncertain modeling on the CrossTask and NIV datasets.

| Metric | Method | CrossTask | | | | NIV | |
|---|---|---|---|---|---|---|---|
| | | $T = 3$ | $T = 4$ | $T = 5$ | $T = 6$ | $T = 3$ | $T = 4$ |
| KL-Div ↓ | Deterministic | 3.12 | 3.88 | 4.39 | 4.04 | 5.40 | **5.29** |
| | Noise | 2.75 | 3.16 | 4.37 | 4.74 | 5.36 | 6.03 |
| | Ours | **2.66** | **2.81** | **2.12** | **1.97** | **4.65** | 5.47 |
| NLL ↓ | Deterministic | 3.70 | 4.45 | 4.98 | 5.34 | 5.48 | **5.42** |
| | Noise | 3.33 | 4.04 | 4.95 | 5.32 | 5.44 | 6.12 |
| | Ours | **3.24** | **3.69** | **3.22** | **3.27** | **4.74** | 5.56 |
| ModePrec ↑ | Deterministic | 52.76 | 41.13 | 31.46 | 18.65 | 27.77 | 26.48 |
| | Noise | 54.30 | 46.15 | 22.52 | 19.09 | 23.19 | 32.39 |
| | Ours | **56.19** | **47.05** | **32.75** | **22.98** | **30.75** | **35.88** |
| ModeRec ↑ | Deterministic | 31.71 | 20.55 | 18.70 | 4.63 | 26.48 | 21.57 |
| | Noise | 43.92 | 22.35 | 21.53 | 17.53 | 32.39 | 23.75 |
| | Ours | **47.34** | **37.97** | **39.64** | **35.03** | **35.88** | **29.90** |

## 5 CONCLUSION

In this paper, we introduce the Masked Temporal Interpolation Diffusion (MTID) model, specifically designed for procedure planning in instructional videos. Our model employs a latent space temporal interpolation module within a U-Net architecture to capture intermediate states and temporal relationships between actions. By incorporating a task-adaptive masked strategy during both inference and loss calculation, MTID improves the accuracy and consistency of generated action sequences. Extensive experiments across the CrossTask, COIN, and NIV datasets demonstrate that our model consistently outperforms existing methods on key metrics. For future work, we aim to further optimize the memory efficiency of the model to handle larger datasets more effectively. Additionally, refining the mask mechanism to enhance control over intermediate state generation and exploring more diverse interpolation strategies remain promising directions. We also plan to extend the application of the temporal interpolation module to broader procedural learning tasks, including more complex conditional planning scenarios.

## REPRODUCIBILITY STATEMENT

To ensure the reproducibility of our work, we provide general details on the datasets and experimental settings in Section 4.1. Comprehensive information on the model architecture, datasets, and training strategies can be found in Appendix A.

## ACKNOWLEDGEMENT

This work is partially supported by the National Natural Science Foundation of China under Grants 62441232, U21B200217, 62306092, 62476068, and 62236008, and projects ZR2024QF066 and ZR2023QF030 supported by Shandong Provincial Natural Science Foundation.

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

# APPENDIX

## A  IMPLEMENTATION DETAILS

### A.1  MODEL ARCHITECTURE DETAILS

In the first learning stage, we aim to predict the task class label given the observations $\{V_s, V_g\}$. We employ a simple 4-layer transformer model for this task and use cross-entropy loss to train the model by comparing its output with the ground truth task class labels.

The classifier is a neural network based on the transformer architecture. It first embeds the input data through a linear layer, after which the embedded data is processed by multiple stacked transformer encoder layers. The output of the encoder layers is averaged and then passed through a series of fully connected layers with ReLU activation functions. Finally, the processed data is passed through a linear layer to generate the final output. Dropout layers are applied throughout the model to prevent overfitting.

Next, our main model is based on a 3-layer U-Net (Ronneberger et al., 2015), similar to Wang et al. (2023b), but adapted for temporal action prediction. Each layer consists of two residual temporal blocks (He et al., 2016), followed by either downsampling or upsampling. Each residual temporal block includes two convolutional layers, group normalization (Wu & He, 2018), Mish activation (Misra, 2019), and a cross-attention module for feature fusion. Temporal embeddings are generated via a fully connected layer and added to the output of the first convolution. To handle the short planning horizon ($T = \{3, 4, 5, 6\}$), we employ 1D convolutions with a kernel size of 2, stride of 1, and no padding for downsampling/upsampling used by Wang et al. (2023b), instead of the max-pooling approach, ensuring the horizon length remains unchanged. The middle block consists of only two residual temporal blocks.

The input matrix $\hat{x}_n$ is a concatenation of the task class, action sequences, and observation features, with a dimension of $fusion\_dim = C + A + O$, where $C$, $A$, and $O$ represent the number of task classes, action labels, and visual features, respectively. During the downsampling phase, the input is embedded through $[fusion\_dim \rightarrow 256 \rightarrow 512 \rightarrow 1024]$, with the reverse process occurring during the upsampling phase.

The latent space temporal interpolation module consists of three main components: an observation encoder, a latent space interpolator, and transformer encoder blocks. The observation encoder reduces the input dimensionality using two 1D convolutional layers with ReLU activations. The latent

space interpolator generates intermediate features between two encoded representations via linear interpolation, guided by a learnable linear layer initialized with matrix $\tau$. The generated matrix is passed through a Sigmoid function to compute $I_j$. Finally, standard transformer encoder blocks apply attention mechanisms to enhance the temporal relationships in $F_j$, ultimately producing transformed latent features with a shape of $[M, O]$, where $M$ refers to the number of residual temporal blocks in the U-Net.

For the diffusion process, we employ a cosine noise schedule to generate $\{\beta_n\}_{n=1}^N$, which controls the amount of noise added at each step. These values correspond to the variance of the Gaussian noise introduced at each stage of diffusion.

## A.2 DATASET DETAILS

Each video in the dataset is annotated with action labels and their corresponding temporal boundaries, denoted as $\{s_i, e_i\}$ for the $i$-th action, where $s_i$ and $e_i$ represent the start and end times, respectively. The total number of actions in the dataset is denoted as $Num$. We extract step sequences $a_{t:(t+T-1)}$ from the videos, with the horizon $T$ ranging from 3 to 6. Following the method in previous work (Wang et al., 2023b), action sequences $\{[a_t, \ldots, a_{t+T-1}]\}_{t=1}^{Num-T+1}$ are generated by sliding a window of size $T$ over the $Num$ actions. For each sequence, the video clip feature at the start time of action $a_t$ is used as the starting observation $V_s$, and the clip feature at the end time of action $a_{t+T-1}$ is used as the goal state $V_g$. Both clips are 3 seconds in duration. The start and end times of each sequence are rounded to $\lfloor s_t \rfloor$ and $\lceil e_{t+T-1} \rceil$, respectively, with the clip features between these times used as $V_s$ and $V_g$.

For the CrossTask dataset, we consider two types of pre-extracted features: (1) the 3200-dimensional features provided by the dataset, which combine I3D, ResNet-152, and audio VGG features (Carreira & Zisserman, 2017; He et al., 2016; Hershey et al., 2017), and (2) features extracted using an encoder trained on the HowTo100M dataset (Miech et al., 2019), as used in (Wang et al., 2023b). We utilize the latter due to its smaller size. For the COIN and NIV datasets, we also use HowTo100M features (Wang et al., 2023b) to maintain consistency and ensure fair comparison.

## A.3 DETAILS OF METRICS

Previous works (Chang et al., 2020; Bi et al., 2021; Sun et al., 2022) computed the mIoU metric over mini-batches, averaging the results across the batch size. However, this method introduces variability depending on the batch size. For instance, if the batch size equals the entire dataset, all predicted actions may be considered correct. In contrast, using a batch size of one penalizes any mismatch between predicted and ground-truth sequences. To address this issue, we follow Wang et al. (2023b) by standardizing mIoU calculation, computing it for each individual sequence and then averaging the results, effectively treating the batch size as one. However, this approach may result in our mIoU scores being lower than those reported by others.

## A.4 TRAINING DETAILS

Following Wang et al. (2023b), we employ a linear warm-up strategy to train our model, with specific protocols adjusted for different datasets. For the CrossTask dataset, we set the diffusion steps to 250 and train for 20,000 steps. The learning rate is linearly increased to $5 \times 10^{-4}$ over the first 3,333 steps, then halved at steps 8,333, 13,333, and 18,333. For the NIV dataset, with 50 diffusion steps, training lasts for 5,000 steps. The learning rate ramps up to $3 \times 10^{-4}$ over the first 1,000 steps and is reduced by 50% at steps 2,666 and 4,332. In the larger COIN dataset, we use 300 diffusion steps and train for 30,000 steps. The learning rate increases to $1 \times 10^{-5}$ in the first 5,000 steps and is halved at steps 12,500, 20,000, and 27,500, stabilizing at $2.5 \times 10^{-6}$ for the remaining steps. Training is performed using ADAM (Kingma, 2014) on 8 NVIDIA RTX 3090 GPUs.

## A.5 DETAILS OF UNCERTAINTY MODELING

In the main paper, we investigate the probabilistic modeling capability of our model on the CrossTask and COIN datasets, demonstrating that our diffusion-based model can generate both di-

verse and accurate plans. Here, we provide additional details, results, and visualizations to further illustrate how our model handles uncertainty in procedure planning.

**Additional Results on COIN.** Results on the COIN dataset are presented in Table 8. On the COIN dataset, our model underperforms relative to the Deterministic baseline. We attribute this to the shorter action sequences, where reduced uncertainty is more advantageous but less critical for long-horizon procedural planning.

**Visualizations for Uncertainty Modeling.** In Figures 7a to 7d, we present visualizations of various plans with the same start and goal observations, generated by our masked temporal interpolation diffusion model on CrossTask for different prediction horizons. We have observed that some results contain repeated actions, which is due to the probabilistic nature of our model's prediction method, making repeated action predictions unavoidable. The top five predicted logits for the actions are passed through a softmax function, and the action with the highest probability is selected to form the prediction figures.

**Details of Evaluating Uncertainty Modeling.** For the Deterministic baseline, we sample once to obtain the plan, as the result is fixed when the observations and task class conditions are given. For the Noise baseline and our diffusion-based model, we sample 1,500 action sequences to calculate the uncertainty metrics. To efficiently perform this process, we apply the DDIM (Song et al., 2021) sampling method to our model, enabling each sampling process to be completed in 10 steps. This accelerates sampling by 20 times for CrossTask and COIN, and by 5 times for NIV. It is important to note that multiple sampling is only required when evaluating probabilistic modeling—our model can generate a good plan with just a single sample.

Table 8: The results of uncertain modeling on the COIN dataset.

| Metric | Method | $T = 3$ | $T = 4$ |
|---|---|---|---|
| KL-Div ↓ | Deterministic | **4.47** | **4.40** |
| | Noise | 5.12 | 4.88 |
| | Ours | 4.74 | 4.47 |
| NLL ↓ | Deterministic | **5.42** | **5.81** |
| | Noise | 6.07 | 6.28 |
| | Ours | 5.69 | 5.87 |
| ModePrec ↑ | Deterministic | **34.04** | **32.47** |
| | Noise | 23.16 | 22.18 |
| | Ours | 28.83 | 26.91 |
| ModeRec ↑ | Deterministic | **27.41** | **20.88** |
| | Noise | 21.06 | 15.24 |
| | Ours | 23.27 | 18.14 |

## B    BASELINE METHODS

In this section, we describe the baseline methods used in our study.

- **Random Selection.** This method randomly selects actions from the available action space within the dataset to generate plans.

- **Retrieval-Based Approach.** Given the observations $\{V_s, V_g\}$, this method retrieves the nearest neighbor by minimizing the visual feature distance within the training dataset. The action sequence associated with the retrieved neighbor is then used as the plan.

- **WLT DO (Ehsani et al., 2018).** This method employs a recurrent neural network (RNN) to predict action steps based on the provided observation pairs.

- **UAAA (Abu Farha & Gall, 2019).** UAAA is a two-stage approach that uses an RNN-HMM model to predict action steps in an auto-regressive manner.

- **UPN (Srinivas et al., 2018).** UPN is a path planning algorithm for physical environments that learns a plannable representation to generate predictions. To produce discrete action steps, a softmax layer is appended to the model's output, as described in (Chang et al., 2020).

- **DDN (Chang et al., 2020).** DDN is an auto-regressive framework with two branches designed to learn an abstract representation of action steps and predict transitions in the feature space.

- **PlaTe (Sun et al., 2022).** PlaTe extends DDN by incorporating transformer modules into its two branches for prediction tasks. PlaTe uses a different evaluation protocol compared to other models.

- **Ext-GAIL (Bi et al., 2021).** Ext-GAIL addresses procedure planning using reinforcement learning. Unlike our approach, Ext-GAIL divides the planning problem into two stages:

the first provides long-horizon information, which is then used by the second stage. In contrast, our approach derives sampling conditions directly.

- **P³IV (Zhao et al., 2022).** P³IV is a transformer-based, single-branch model that incorporates a learnable memory bank and an additional generative adversarial framework. Similar to our model, P³IV predicts all action steps simultaneously during inference.

- **EGPP (Wang et al., 2023a).** EGPP proposes an event-guided paradigm for procedure planning, where events are inferred from observed visual states to guide the prediction of intermediate actions. The E3P model uses event-aware prompting and Action Relation Mining to improve action prediction accuracy, significantly outperforming existing methods in experiments.

- **PDPP (Wang et al., 2023b).** PDPP is a two-branch framework that models temporal dependencies and action transitions using a diffusion process. Like our model, PDPP predicts all actions simultaneously, refining predictions over multiple stages to enhance logical consistency during inference.

- **KEPP (Nagasinghe et al., 2024).** KEPP is a knowledge-enhanced procedure planning system that leverages a probabilistic procedural knowledge graph (P2KG) learned from training plans. This graph acts as a "textbook" to guide step sequencing in instructional videos. KEPP predicts all action steps simultaneously with minimal supervision, achieving leading performance.

- **SCHEMA (Niu et al., 2024).** SCHEMA focuses on procedure planning by learning state transitions. It employs a transformer-based architecture with cross-modal contrastive learning to align visual inputs with text-based state descriptions. By tracking intermediate states, SCHEMA predicts future actions using a large language model to capture temporal dependencies and logical transitions, improving action planning in instructional videos.

## C ADDITIONAL ABLATION STUDIES

**Model Size.** Due to the large size of the COIN dataset, we adjust the model size by modifying the U-Net architecture. As shown in Table 9, increasing the model size results in higher scores for the COIN dataset. We believe that optimizing the model to be more memory-efficient could further improve performance, which we plan to explore in future work. In Table 9, increasing the size to 512 does not improve the scores on CrossTask. We believe this suggests overfitting, indicating that a model size of 256 is sufficient for this task.

Table 9: Ablation study on the role of model size on COIN and CrossTask datasets.

| Dataset | Size | $T = 3$ | | | $T = 4$ | | |
|---------|------|------|------|------|------|------|------|
| | | SR↑ | mAcc↑ | mIoU↑ | SR↑ | mAcc↑ | mIoU↑ |
| COIN | 128 | 23.01 | 45.44 | 51.93 | 19.69 | 45.32 | 55.06 |
| | 256 | 28.84 | 50.44 | 57.86 | 21.64 | 48.06 | 59.52 |
| | 512 | **30.90** | **52.17** | **59.58** | **23.10** | **49.71** | **60.78** |
| CrossTask | 128 | 23.01 | 45.44 | 51.93 | 19.69 | 45.32 | 55.06 |
| | 256 | **40.45** | **67.19** | **69.17** | **24.76** | **60.69** | **67.67** |
| | 512 | 37.94 | 65.16 | 67.43 | 21.97 | 58.30 | 66.15 |

**Components of Observation Encoder.** Table 10 presents the impact of different encoder components. Based on this ablation study, the optimal model consists of two 1D convolution layers with ReLU activation, which achieves the best balance between depth and activation, resulting in the highest scores across all metrics. Adding more layers does not consistently improve performance, and activation functions like ReLU play a key role in enhancing model effectiveness. We believe that ReLU introduces non-linearity, enabling the network to capture temporal latent features more effectively. Moreover, by setting negative values to zero, ReLU promotes sparse activation, which may aid in the extraction and construction of latent features.

Table 10: Ablation study on the observation encoder components ($T = 3$, CrossTask).

| Models | SR↑ | mAcc↑ | mIoU↑ |
|---|---|---|---|
| 1 1d conv. layer w/ ReLU | 39.71 | 66.91 | 69.05 |
| 2 1d conv. layers w/ ReLU (Ours) | **40.45** | **67.19** | **69.17** |
| 3 1d conv. layers w/ ReLU | 36.04 | 64.41 | 66.55 |
| 1 1d conv. layer w/o ReLU | 39.32 | 66.70 | 68.91 |
| 2 1d conv. layers w/o ReLU | 39.38 | 66.73 | 68.65 |
| 3 1d conv. layers w/o ReLU | 37.77 | 65.06 | 67.61 |

Table 11: Ablation study on the role of classifier type on CrossTask dataset.

| Models | $T = 3$ | | | $T = 4$ | | | $T=5$ | $T = 6$ |
|---|---|---|---|---|---|---|---|---|
| | SR↑ | mAcc↑ | mIoU↑ | SR↑ | mAcc↑ | mIoU↑ | SR↑ | SR↑ |
| PDPP (Res-MLP) | 37.2 | 55.35 | 66.57 | 21.48 | 57.82 | 65.13 | 13.58 | 8.47 |
| PDPP (Transformer) | 39.08 | 66.32 | 68.47 | 22.48 | **60.72** | 66.13 | 13.77 | 8.63 |
| MTID (Transformer) | **40.45** | **67.19** | **69.17** | **24.76** | 60.69 | **67.67** | **15.26** | **10.30** |

**Transformer Classifier Type.** We conduct ablation studies on the CrossTask and COIN datasets to evaluate the impact of our transformer-based classifier. As shown in Tables 11 and 12, the inclusion of the transformer-based classifier significantly boosts the performance of PDPP. Although the improvements are modest for longer horizons, this highlights the effectiveness of our temporal interpolation module on CrossTask compared to PDPP with the same transformer classifier. However, the classifier's performance may also limit further improvements. Additionally, we observe that even with incorrect task class labels during supervision, the model still achieves strong scores, demonstrating its robustness, fault tolerance, and error correction capabilities.

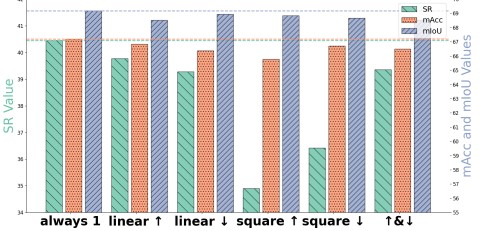
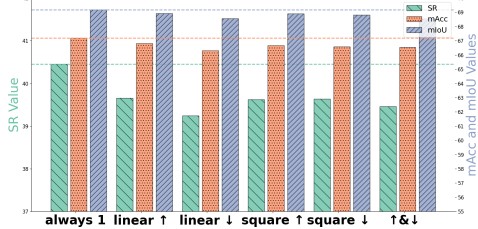

(a) Ablation studies on different complex initialization methods with max value 6.

(b) Ablation studies on different complex initialization methods with max value 1.

Figure 5: Ablation studies for interpolation strategy. Note: "always 1" indicates that $\tau$ is initialized to '1'; "linear ↑" denotes that the values in the matrix $\tau$ increase linearly, with the first column initialized to '1' and the last column set to '6' in Figure 5a, with a gradual linear increase in between, and from '0' to '1' in Figure 5b; "linear ↓" represents the reverse process. "square ↑" indicates that the value of $\tau$ increases following a square progression. "↑ & ↓" refers to a variation similar to our gradient loss weights, where the value first increases linearly and then decreases.

**More Interpolation Strategies.** We experimented with both linear and non-linear strategies, as shown in Figure 5. In Figure 5a, we found that using a maximum value of '6' led to poor results, particularly for the "square ↑" and "square ↓" strategies, indicating a significant deviation from the desired final value. When we reduced the maximum value to '1', the results still remained unsatisfactory, suggesting that the final value of $\tau$ consistently converged around '1', resulting in sub-optimal performance.

**Number of Transformer Encoder Layers.** Figure 6a shows the scores of three metrics across different numbers of layers in the transformer encoder blocks. The results indicate that using fewer layers (1 or 2) results in a significant drop in performance, with SR being the most adversely affected. As the number of layers increases, the metrics stabilize, with notable improvements, especially in

Table 12: Ablation study on the role of classifier type on COIN dataset.

| Models | T = 3 | | | T = 4 | | |
|---|---|---|---|---|---|---|
| | SR↑ | mAcc↑ | mIoU↑ | SR↑ | mAcc↑ | mIoU↑ |
| PDPP (Res-MLP) | 21.33 | 45.62 | 51.82 | 14.41 | 44.10 | 51.39 |
| PDPP (Transformer) | 24.02 | 48.03 | 55.21 | 17.36 | 46.12 | 55.82 |
| MTID (Transformer) | **28.84** | **50.44** | **57.86** | **21.64** | **48.06** | **59.52** |

SR, which shows a significant positive shift at 6 and 7 layers. In contrast, mAcc and mIoU show more subtle variations, with slight positive changes as the number of layers increases, reflecting a steady trend. These results suggest that an optimal configuration of 6 or 7 layers delivers the best overall performance.

Table 13: Performance comparison of T and M.

| Method | SR↑ | mAcc↑ | mIoU↑ |
|---|---|---|---|
| T | 38.64 | 66.13 | 68.05 |
| M | **40.45** | **67.19** | **69.17** |

Table 14: Ablation studies on different components on CrossTask. This ablation study effectively demonstrates the predictive capability of our method. Note: The results of ID 1 are from PDPP.

| ID | Interpolation Module | Mask Projection | Proximity Loss | SR↑ | mAcc↑ | mIoU↑ |
|---|---|---|---|---|---|---|
| 1 | | | | 37.20 | 64.67 | 66.57 |
| 2 | ✓ | | | 39.03 | 66.49 | 68.26 |
| 3 | | ✓ | | 38.88 | 66.36 | 68.35 |
| 4 | | | ✓ | 38.57 | 66.02 | 68.17 |
| 5 | ✓ | ✓ | | 39.64 | 66.74 | 68.77 |
| 6 | ✓ | | ✓ | 39.71 | 66.65 | 68.83 |
| 7 | | ✓ | ✓ | 39.17 | 66.49 | 68.38 |
| 8 | ✓ | ✓ | ✓ | **40.45** | **67.19** | **69.17** |

**Ablation for Our Different Methods.** Table 14 presents the effects of our proposed methods. The results demonstrate that each component significantly enhances the model's performance.

## D    MORE ANALYSIS FOR METHODS

**More Explanation for $M$.** Our MTID diffusion model takes as input a matrix containing action sequences with T timesteps and is based on U-Net, which contains M residual temporal blocks in the downsampling, upsampling, and middle layers for directly diffusing and generating T intermediate target actions. To ensure that each intermediate layer contains valid auxiliary information, our Latent Space Temporal Interpolation Module needs to generate M intermediate auxiliary features. Subsequently, we apply cross-attention in residual temporal blocks across the M interpolated features and the entire input matrix rather than individual timesteps, enabling better temporal integration. We also conducted experiments to demonstrate the effect of M. Our results in Table 13 showed that using interpolated features only for T steps led to suboptimal performance. This also supports our decision to use interpolated features across all M modules.

**More Explanation for Weighted Gradient Loss.** In PDPP, for handling the loss, a coefficient $w$ with a value of 10 is multiplied at the beginning and end positions. The paper believes that both sides are more important because they are the most related actions for the given observations. Experiments have shown that this approach is indeed effective. However, in my opinion, it doesn't better conform to the pattern of action accuracy. From my observations, I found that the accuracy tends to decrease for actions closer to the middle, but not identical in the middle as shown in Figure 6b. This is because the supervision from real visual features is stronger on both ends, while, as we move toward

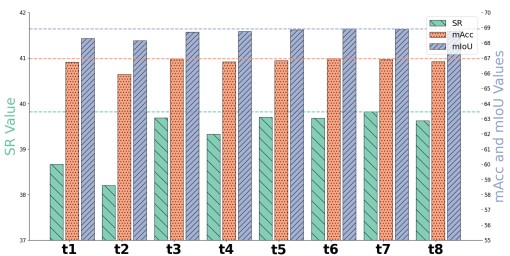

(a) Ablation studies on the number of transformer encoder block layers.

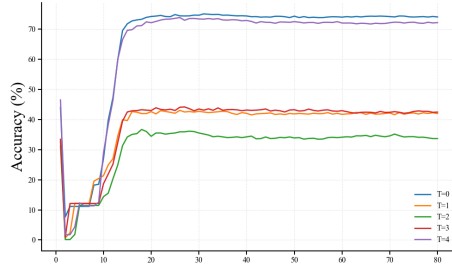

(b) Accuracy changes of different actions at various time steps as epochs progress when T=5.

Figure 6: Combined ablation studies of different coefficients. Note: "t1" refers to the number of transformer encoder layers in Figure 6a.

the middle, there is actually less available real information. Inspired by this, I designed a loss function that imposes stricter supervision constraints on both ends while relaxing the constraints in the middle, achieving better results.

**Upper Bound of Visual Features Supervision.** The comparison presented in Table 15 reveals that results vary depending on dataset characteristics, particularly size, task types, and average action sequence lengths. To explain this, we categorize our interpolated features into two parts: simple memory and hard temporal relationships. For instance, COIN, which has the largest dataset size but the shortest sequences, demonstrates that interpolated features excel in tasks focused on simple memory. In contrast, NIV, being the smallest dataset with the longest sequences, shows comparable performance between real and interpolated features. Meanwhile, CrossTask, characterized by large size and long sequences, exhibits a significant performance gap favoring real features. These findings highlight a trade-off where interpolated features perform well in simpler datasets but struggle with complex temporal relationships in larger, more diverse datasets. This underscores the necessity for improved interpolation methods to effectively manage complex, temporally diverse datasets in future research.

Table 15: Combined results for CrossTask, COIN, and NIV datasets with interpolated features and original real features.

| Dataset | Method | $T = 3$ | | | $T = 4$ | | | $T=5$ | $T = 6$ |
| | | SR↑ | mAcc↑ | mIoU↑ | SR↑ | mAcc↑ | mIoU↑ | SR↑ | SR↑ |
|---------|--------|------|-------|-------|------|-------|-------|------|------|
| CrossTask | Interpolated | 40.45 | 67.19 | 69.17 | 24.76 | 60.69 | 67.67 | 15.26 | 10.30 |
| | Real | **49.05** | **73.62** | **73.23** | **36.55** | **70.42** | **72.09** | **24.88** | **24.02** |
| COIN | Interpolated | **30.90** | **52.17** | **59.58** | **23.10** | **49.71** | **60.78** | – | – |
| | Real | 27.07 | 49.07 | 57.53 | 20.01 | 47.35 | 58.24 | – | – |
| NIV | Interpolated | 29.63 | 48.02 | **56.49** | **25.76** | 46.62 | 58.50 | – | – |
| | Real | **32.59** | **50.25** | 56.40 | 24.02 | **48.36** | **58.92** | – | – |

**Comparison with PDPP.** The results on COIN and NIV under the PDPP settings, as presented in Table 16, indicate that our performance on NIV is slightly lower due to two main factors. First, the dataset size of NIV is significantly smaller than that of CrossTask and COIN, which leads to the model excessively learning detailed patterns from the training data and consequently reducing its generalization ability. Second, there are differences in experimental settings: PDPP defines states as the window between start and end times, while KEPP uses a 2-second window around start and end times. This difference allows PDPP to access more step information, particularly for short-term actions, which may weaken the impact of our interpolation feature supplementation. Despite these challenges with NIV under PDPP settings, our model demonstrates strong capabilities on the larger CrossTask and COIN datasets, highlighting its effectiveness in temporal logic and memory utilization.

Table 16: Comparisons between PDPP and MTID under the setting of PDPP.

| Models | COIN | | | | | | NIV | | | | | |
| | $T = 3$ | | | $T = 4$ | | | $T = 3$ | | | $T = 4$ | | |
| | SR↑ | mAcc↑ | mIoU↑ | SR↑ | mAcc↑ | mIoU↑ | SR↑ | mAcc↑ | mIoU↑ | SR↑ | mAcc↑ | mIoU↑ |
|---|---|---|---|---|---|---|---|---|---|---|---|---|
| PDPP | 21.33 | 45.62 | 51.82 | 14.41 | 44.10 | 51.39 | **30.20** | **48.45** | **57.28** | **26.67** | **46.89** | **59.45** |
| MTID | **30.90** | **52.17** | **59.58** | **23.10** | **49.71** | **60.78** | 29.63 | 48.02 | 56.49 | 25.76 | 46.62 | 58.50 |

# E  FURTHER DISCUSSIONS

**Limitations.** The limitations of our method are as follows. First, while the logical consistency between the actions generated by our model is generally strong, there is no guarantee of perfect alignment with the task. Mismatches were observed during the experiments, which is a common issue in procedure planning models. This challenge arises because the labels for multi-task and multi-action tasks in the dataset are replaced by data IDs, which may lead to issues with numerical calculations.

**Comparison Across Supervision Strategies and Mid-State Handling.** Our MTID model introduces several innovations that set it apart in terms of how it handles supervision and mid-state action prediction: (1) *Supervision Approach: Weak vs. Full Supervision*: DDN, PlaTe, and Ext-GAIL rely on fully supervised learning, requiring extensive annotations to model temporal dynamics. In contrast, MTID uses a weakly supervised approach with a latent space temporal interpolation module, capturing mid-state information without detailed annotations. Its diffusion process and latent interpolation offer finer-grained supervision for intermediate steps, outperforming Ext-GAIL and DDN in long-term predictions. (2) *Intermediate State Supervision and Logical Structure*: PDPP uses task labels to bypass intermediate state supervision, and Skip-Plan reduces uncertainty by skipping uncertain intermediate actions. However, both methods struggle to fully capture the logical structure of intermediate steps. MTID addresses this by explicitly supervising mid-state actions through latent space interpolation, ensuring that the generated sequences are both temporally logical and well-aligned with the task requirements. (3) *Handling of External Knowledge and Probabilistic Guidance*: P3IV leverages natural language instructions for weak supervision, while KEPP uses a probabilistic procedural knowledge graph (P2KG) to guide the planning process. While both methods aim to improve action prediction through external guidance, MTID distinguishes itself by focusing on direct mid-state supervision via intermediate latent features from a diffusion model. This approach provides more precise control over action generation, ensuring logical consistency across the entire sequence. (4) *State Representation and Visual Alignment*: SCHEMA relies on large language models (LLMs) to describe and align state changes with visual observations, focusing on high-level state transitions. MTID, in contrast, directly uses mid-state supervision through latent space temporal interpolation, which improves visual-level supervision and enhances temporal reasoning, resulting in more accurate action sequence predictions.

**Generalization Capabilities.** Our MTID model demonstrates strong generalization across variations in action steps, object states, and environmental conditions. For action step variations, the model was evaluated with sequences of different lengths, ranging from 3 to 6 steps. The results consistently showed that MTID outperforms state-of-the-art models, leveraging its latent space temporal interpolation to capture temporal relationships across various step lengths. In terms of object states and environmental contexts, the benchmark datasets used for evaluation cover a wide range of topics, such as cooking, housework, and car maintenance, featuring diverse objects like fruits, drinks, and household items. For instance, the CrossTask dataset includes 105 step types across 18 tasks, while the COIN dataset features 778 step types over 180 tasks. These tests highlight MTID's ability to generalize effectively, capturing the nuances of varying object states and environmental conditions, due to its robust interpolation and diffusion framework.

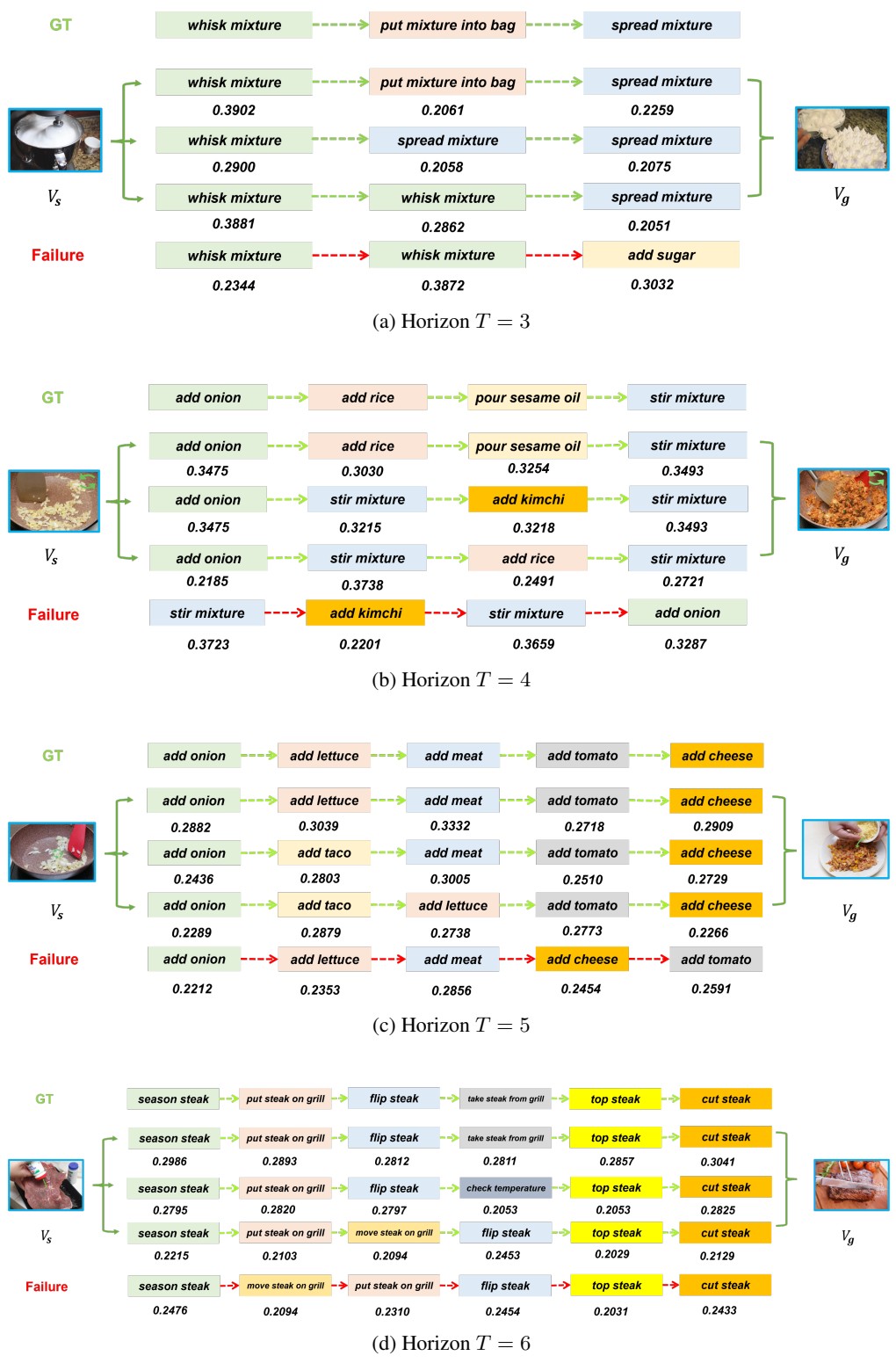

Figure 7: Visualization of diverse plans produced by our model with different horizons. Note: each figure includes images depicting the start and goal observations, the first row labeled "GT" showing the ground truth actions, the last row labeled "Failure" illustrating a plan that does not achieve the goal, and the middle rows displaying multiple reasonable plans produced by our model. These decimals represent the probability values obtained from action prediction.

