# OpenReview forum: "Masked Temporal Interpolation Diffusion for Procedure Planning in Instructional Videos"
_ICLR.cc/2025/Conference — ICLR 2025 Poster_

### Official Review · Reviewer_DGZV · 2024-10-20

**Soundness:** 3
**Presentation:** 3
**Contribution:** 2
**Rating:** 6
**Confidence:** 5

**Summary:**

This paper proposes a Masked Temporal Interpolation Diffusion model to tackle the problem of procedure planning in instructional videos, including a latent space temporal logical interpolation module. The interpolated mid-state visual feature is used to guide the intermediate process within the diffusion model. The MTID achieves SOTA performance on several datasets.

**Strengths:**

1. Based on previous works, this paper additionally uses the interpolated mid-state latent visual feature to supervise the model.
2. The proposed MTID model achieves SOTA performance on several datasets.

**Weaknesses:**

1. The core concept is using the interpolated visual feature to provide visual-level mid-state supervision. However, the motivation of this operation is unclear. Using linear interpolation to interpolate the mid-state visual feature based on the start and goal visual feature is challenging. Additionally, an experiment that directly uses the visual features of mid-state for supervision (like previous works DDN) should be added, which can be seen as an upper bound.
2. In this paper, the task classifier uses a transformer, howerer,  in previous papers (EGPP, PDPP), a MLP is enough for this simple classification task.
3. The masked projection introduces too much prior (i.e., the action space of each task).
4. More analysis should be provided about the Task-Adaptive Masked Proximity Loss ablation, (MSE+GW achieves 15.1, while MSE+W+GW+M achieves 15.26). What's the weighted gradient loss, the authors should give more detail in the Method part.
5. The novelty of the proposed method is limited compared with previous work PDPP.

**Questions:**

Refer to the weaknesses.

---

> ### Author Response · Authors · 2024-11-22
> **Response to reviewer DGZV.**
>
> Dear Reviewer DGZV,
>
> Thank you for your detailed comments and suggestions. We tried our best to address all the concerns and questions, and update the main paper and appendix(marked red) in the new version. Please let us know if you have any further concerns or questions to discuss.
>
> Best,
>
> Paper 1751 Authors
>
> ___
>
> **W1. About the motivation of the interpolation and more experiments for upper bound.**
>
> **A1.**
>
> **Motivation&Challenge**:
> - **Task**: Our method addresses a fundamental challenge in instructional video procedure planning - predicting intermediate actions when only given start and end visual information. Since we lack direct supervision for intermediate states during training and inference, developing an effective approach to _reconstruct_ these missing visual features becomes crucial for accurate prediction.
> - **Supervision level**: While existing approaches are constrained by text-level supervision that offers only limited insights into visual state transitions, our method advances beyond this by incorporating _visual-level_ supervision from start and end observations. However, we face a critical challenge - the absence of real intermediate visual features during the denoising process. This limitation motivates our novel interpolation approach that reconstructs these missing visual states, enabling us to model the temporal progression in significantly more detail and comprehensiveness compared to methods relying solely on text supervision.
> - **Interpolation**: We propose interpolation as an elegant and effective solution to generate intermediate visual features, addressing the absence of real intermediate visual features especially during inference. Visual features inherently possess continuity and logical coherence, allowing interpolation to naturally preserve the holistic semantic information while introducing rich and diverse details. Furthermore, the mathematical simplicity and computational efficiency of interpolation make it an ideal choice that seamlessly integrates into our framework.
>
>
> However, this task remains **challenging**. As shown in Table 6 (ablation experiments for our latent space temporal logical interpolation module), using only an interpolator does not achieve the best performance. To address this, we have implemented several enhancements:
>
> - We added an _observation encoder_ before the interpolator to extract latent features and filter out noise.
> - We incorporated _transformer encoder blocks_ to enhance logical relationships between features.
>
> These additions have significantly improved our model's performance, leading to the best results among compared methods.
>
> **Upper Bound**: As suggested, we have conducted additional experiments using real visual features as supervision to test the upper bound of our method. The outcomes vary based on dataset characteristics (size, especially the kinds of tasks and actions, and average action sequence lengths). To explain this phenomenon, we divide our interpolated features into two parts: _simple memory_ and _hard temporal logical relationships_.
>
> For COIN, with its largest size but shortest sequences, interpolated features excel in simple memory-focused tasks. NIV, the smallest but with longest sequences, shows comparable performance between real and interpolated features. CrossTask, being large with long sequences, reveals a significant performance gap favoring real features.
>
> These findings highlight a trade-off: interpolated features perform well in simpler datasets but struggle with complex temporal relationships in larger, more diverse datasets. This underscores the need for improved interpolation methods to effectively handle complex, temporally diverse datasets in future work.
>
> Table 1: Upper bound of visual features supervision.
> | Dataset | Method | SR↑ (T=3) | mAcc↑ (T=3) | mIoU↑ (T=3) | SR↑ (T=4) | mAcc↑ (T=4) | mIoU↑ (T=4) | SR↑ (T=5) | SR↑ (T=6) |
> |---------|---------|-----------|-------------|-------------|-----------|-------------|-------------|-----------|-----------|
> | CrossTask | Interpolated | 40.45 | 67.19 | 69.17 | 24.76 | 60.69 | 67.67 | 15.26 | 10.30 |
> | | Real | **49.05** | **73.62** | **73.23** | **36.55** | **70.42** | **72.09** | **24.88** | **24.02** |
> | COIN | Interpolated | **30.90** | **52.17** | **59.58** | **23.10** | **49.71** | **60.78** | - | - |
> | | Real | 27.07 | 49.07 | 57.53 | 20.01 | 47.35 | 58.24 | - | - |
> | NIV | Interpolated | 29.63 | 48.02 | **56.49** | **25.76** | 46.62 | 58.50 | - | - |
> | | Real | **32.59** | **50.25** | 56.40 | 24.02 | **48.36** | **58.92** | - | - |
>
> Revision. In the revision, we have added the discussion of upper bound of visual features supervision in Appendix D under the section "Upper Bound of Visual Features Supervision" at line 1000.

---

> ### Author Response · Authors · 2024-11-22
> **Response to reviewer DGZV.(2)**
>
> **W2. A MLP is enough for this simple classification task.**
>
> **A2.** For long instructional videos, MLP classifiers are too simple to model complex spatial-temporal relationships between actions to obtain accurate task labels. To address this limitation, we propose transformer-based classifiers, which can better capture contextual relationships between classes through their self-attention mechanisms and long-term temporal relationship modeling capabilities. This is particularly important since classification accuracy is a crucial factor in the overall performance of our method.
>
> To validate this, we conducted comprehensive ablation studies on classifier types in Appendix C (see "Transformer Classifier Type" section at line 863). Our experimental results (two tables below) show that using a better-performing transformer classifier leads to significant improvements in the overall results themselves under the same conditions.
>
> Table 2: Classification results on CrossTask.
> | Models | T=3 | T=4 | T=5 | T=6 |
> |--------|-----|-----|-----|-----|
> | Res-MLP(PDPP) | 92.43 | 92.98 | 93.39 | 93.20 |
> | Transformer(Ours) | **93.67** | **94.03** | **94.02** | **94.26** |
>
> Table 3: Performance comparison on CrossTask.
> | Model| SR↑ | mAcc↑ | mIoU↑ |
> |------|-----|--------|--------|
> | PDPP (Res-MLP) | 37.2 | 55.35 | 66.57 |
> | PDPP (Transformer) | _39.08_ | _66.32_ | _68.47_ |
> | MTID (Transformer) | **40.45** | **67.19** | **69.17** |
>
> Moreover, compared to PDPP's MLP and our transformer-based classifier, here is the resource comparison on CrossTask (with GeForce RTX 4090):
>
> Table 4: Resource comparison on CrossTask.
> | Model Type    | Memory Usage | Training Time |
> |--------------|--------------|---------------|
> | PDPP (MLP)   | 1237M       | 21min         |
> | Ours (Transformer) | 1637M | 25min   |
>
> Given the significant performance improvements shown in Table 2&3, we believe this slight increase in computational resources is a worthwhile **trade-off**.
>
> ___
>
>
> **W3. The masked projection introduces too much prior.**
>
> **A3.** The masked projection serves as a training-time constraint that helps guide and restrict the action space during initialization. Rather than enforcing a rigid mask prior at inference time, it acts as a soft guidance mechanism during model training.
>
> Our experimental results demonstrate the effectiveness of this approach. As shown in the table below, even without utilizing masked projection (MP), our method still outperforms other compared approaches on the CrossTask dataset, highlighting the robustness of our overall methodology.
>
> Table 5: Performance comparison on CrossTask.
> | Method | SR↑ | mAcc↑ |
> |--------|-----|--------|
> | KEPP | 38.12 | _64.74_ |
> | SCHEMA | _38.93_ |63.80 |
> | MTID w/o MP | **39.17** | **66.49** |

---

> ### Author Response · Authors · 2024-11-22
> **Response to reviewer DGZV.(3)**
>
> **W4. Details about the weighted gradient loss and analyze the loss ablation results.**
>
> **A4.**
>
> **Analysis for ablation of loss**: We explain these results (where the improvement from ID 3 to 6 is only 0.16) by noting that our mask not only filters irrelevant actions but also provides weight constraints on intermediate actions. However, since the mask applies equal weights to all intermediate timesteps, its effectiveness is not as strong as GW which applies gradient weights. This explains why using both together (mask and GW) shows improvement but the gain is relatively small due to their partially overlapping effects.
>
> This explanation is supported by evidence from the improvements seen from ID 1 to 3 and ID 1 to 4. Both methods show approximately 2-point improvements: The mask increases performance from 11.89 (ID1) to 13.26 (ID4), while gradient weights improve performance from 11.89 (ID1) to 13.90 (ID2). These results demonstrate that both our mask and gradient weights independently improve model performance effectively. When used together, although the improvement is modest, it is still noticeable.
>
> Table 6: Ablation studies on our loss function. Note: W: Weights on Both Sides, GW: Gradient Weights, M: Mask.
>
> | ID | MSE | GW | M | SR↑ |
> |----|-----|----|----|-----|
> | 1  | ✓   |    |    | 11.89 |
> | 3  | ✓   | ✓  |    | _15.10_ |
> | 4  | ✓   |    | ✓  | 13.26 |
> | 6  | ✓   | ✓  | ✓  | **15.26** |
>
>
> **Explanation for weighted gradient loss**: In our task-adaptive masked proximity loss,
> $$    \mathcal{L} _ {\mathrm{diff}} = \sum _ {t=1}^{T} \sum _ {d=1}^{A} w _ t \cdot m _ {t,d} \cdot (a _ {t,d} - \bar{a} _ {t,d})^2,$$
> $$w_t = w_0 + (1 - w_0) \cdot \frac{\min(t, T - t + 1) - 1}{\lceil T/2 \rceil - 1},$$
> the weight $w_t$ linearly decreases from both ends toward the middle in a slope-like pattern, so that is why we refer to them as gradient weights in our experiments. The gradient weights are combined with MSE to form the weighted gradient loss, emphasizing the gradual change of weights across different timesteps.
>
>
> > **Note:** Weighted Loss and Weighted Gradient Loss are two different weighting strategies, and they can not be used together. Therefore, in ID 6 we only use MSE+GW+M.

---

> ### Author Response · Authors · 2024-11-22
> **Response to reviewer DGZV.(4)**
>
> **W5. The novelty is limited compared with PDPP.**
>
> **A5.** Our method tackles a fundamental challenge in instructional video procedure planning: accurately predicting the sequence of intermediate actions given only the start and end visual states. This complex task demands sophisticated capabilities in both temporal prediction and logical reasoning. While the existing PDPP approach employs a basic U-Net architecture with diffusion-based prediction, it falls short in capturing the intricate relationships between actions. In contrast, our method introduces several innovative components that significantly enhance the modeling of temporal and logical dependencies between actions:
>
> 1. **Supervision-level Innovation**:
>    - Challenge: Existing approaches, particularly PDPP, primarily utilize text-level supervision, which inherently limits their ability to capture the rich visual dynamics and state transitions between actions.
>    - Our Solution: We introduce a novel visual-level supervision paradigm that leverages both start and end state observations. Through our innovative interpolation mechanism, we effectively reconstruct intermediate visual features that would otherwise be missing. This approach enables our model to capture temporal progression with significantly greater fidelity compared to traditional text-based supervision methods.
>
> 2. **Architecture-level Innovations**:
>
>    a) **Latent Space Temporal Logical Interpolation Module**:
>       - Challenge: PDPP's simple U-Net architecture lacks the sophistication needed to effectively model and capture the complex temporal relationships and logical dependencies between sequential actions.
>       - Our Improvement: We propose a novel interpolation module that intelligently reconstructs missing intermediate visual features, significantly enhancing the model's ability to reason about temporal progression and seamlessly integrate features across different timesteps.
>
>    b) **Task-adaptive Masked Projection**:
>       - Challenge: PDPP's design leads to predictions more likely to fall outside the current task's action space.
>       - Our Improvement: We implement an adaptive mechanism to constrain the action space during initialization by masking out irrelevant actions for current task, reducing out-of-scope predictions.
>
> 3. **Loss-level Innovation**:
>    - Challenge: PDPP's loss function causes predictions more likely to fall outside the task's action space and overemphasizes endpoints.
>
>    - Our Improvements:
>
>      a) **Gradient-guided weighting**: Balances supervision across all timesteps while maintaining emphasis on endpoints.
>
>      b) **Task-specific mask mechanism**: Further limits the action space to align with task-specific constraints.
>
> 4. **Efficiency Improvements**:
>    - Challenge: PDPP's DDPM diffusion process is time-consuming.
>    - Our Improvement: We replace it with DDIM, accelerating both training and inference while maintaining prediction quality.
>
> 5. **Task Classification**:
>    - Challenge: PDPP's simple MLP classifier is insufficient for complex task classification.
>    - Our Improvement: We enhance it with a transformer module, improving complex task classification through better modeling of long-term temporal relationships.
>
> These innovations work synergistically to significantly advance the state-of-the-art across benchmarks (CrossTask: +3 points, COIN: +9 points compared to PDPP), demonstrating substantial improvements over PDPP's framework.

---

> ### Author Response · Authors · 2024-11-26
> **Response to reviewer DGZV.(5)**
>
> Dear reviewer DGZV,
>
> Thank you for your thoughtful comments. We understand that there may still be some ambiguity regarding our core concept. In our revised manuscript, we have made substantial efforts to clarify these points and better highlight the novel aspects of our work. We hope that our explanations have addressed your concerns, but we remain happy to provide any further clarifications if needed.
>
> Please feel free to share any additional feedback. We greatly value your continued engagement and the opportunity to improve our work.
>
> Best,
>
> Paper 1751 Authors

---

> > ### Comment · Reviewer_DGZV · 2024-11-27
> > **Response to the authors**
> >
> > Thank you for your response, which addresses some of my questions regarding the motivation and the upper bound. However, the actual effect of the interpolation operation remains unclear. Compared to the motivation, linear interpolation appears overly simplistic, and I doubt whether this approach can meet the goal. Additional experiments should be conducted to substantiate this point. Overall, I have increased my score to 6.

---

> > > ### Author Response · Authors · 2024-12-03
> > > **Response to reviewer DGZV.(6)**
> > >
> > > We sincerely appreciate the reviewer's recognition of our work and the increased score. Here's our detailed response for simplicity of linear interpolation:
> > >
> > > 1. Interpolation Strategy: We adopt learnable linear interpolation to generate intermediate features between key frames. Although this approach is straightforward, it performs effectively in our system by creating smooth transitions between visual features. As shown in Figure 4(b)(line 436), the model performs poorly without interpolation, demonstrating the importance of the interpolation operation. In Figure 4(c)(line 446), applying interpolation twice on the features leads to deteriorated results, indicating that repeated interpolation causes information loss from the original features. In Figure 5(a) and (b)(line 890), we compare various interpolation strategies, including constant, linear, and quadratic interpolation, along with different initialization directions (increasing and decreasing). The quadratic interpolation shows no improvement in performance, suggesting that more complex interpolation methods are unnecessary. Moreover, the results indicate that the interpolation effectiveness remains consistent regardless of whether the interpolation values initially increase or decrease, showing its independence from action-related gradient changes. Furthermore, the learnable design enables more flexible and effective feature transitions that can adapt to different actions.
> > >
> > > 2. Feature Diversity and Integration: In our task, each action corresponds to unique visual features. The interpolation strategy helps create diverse and distinct features during training, which is crucial for model performance, as demonstrated by Super SloMo[1]. The effectiveness of our Integration approach stems from the synergistic combination of multiple components. We employ cross-attention mechanisms similar to IP-Adapter[2] for visual feature fusion, which proves more effective than direct concatenation. Our ablation studies in Table 6 (line 465) demonstrate that the holistic integration of architectural design choices and feature fusion mechanisms leads to strong results.
> > >
> > > 3. Computational Efficiency: Linear interpolation offers practical advantages in terms of implementation simplicity and computational efficiency, making it a reasonable choice given its role in the broader architecture.
> > >
> > > In summary, exploring more sophisticated and effective interpolation strategies is an promising direction for our future research.
> > >
> > > [1] Jiang, H., Sun, D., Jampani, V., Yang, M. H., Learned-Miller, E., & Kautz, J. (2018). Super slomo: High quality estimation of multiple intermediate frames for video interpolation. In Proceedings of the IEEE conference on computer vision and pattern recognition (pp. 9000-9008).
> > >
> > > [2] Ye, H., Zhang, J., Liu, S., Han, X., & Yang, W. (2023). Ip-adapter: Text compatible image prompt adapter for text-to-image diffusion models. arXiv preprint arXiv:2308.06721.

---

### Official Review · Reviewer_qUxm · 2024-11-02

**Soundness:** 3
**Presentation:** 4
**Contribution:** 3
**Rating:** 8
**Confidence:** 4

**Summary:**

This paper introduces the Masked Temporal Interpolation Diffusion model to improve procedure planning in instructional videos. MTID integrates a latent space temporal logical interpolation module into a diffusion framework, generating richer visual supervision for intermediate states. The model is evaluated across benchmarking datasets and has been shown to achieve promising results.

**Strengths:**

The proposed MTID model is a novel application of diffusion models for procedural planning, introducing an innovative interpolation mechanism to enrich visual supervision, which is rarely addressed in the field.

The paper provides strong experimental evidence demonstrating that MTID outperforms existing models on multiple benchmark datasets. The results on CrossTask, COIN, and NIV datasets suggest significant improvements in most evaluation metrics. Plus the ablation studies are detailed and informative.

**Weaknesses:**

The model's architecture, especially the use of latent space interpolation combined with diffusion processes, may be difficult for readers to grasp fully. More intuitive visualizations or a simplified explanation could aid in better understanding.

**Questions:**

Does the task-adaptive masked proximity loss sufficiently balance intermediate state supervision with endpoint accuracy, or are there alternative loss functions that could enhance performance?

The classification results in NIV datasets seems to be way better than that of the COIN dataset. However, as shown in Table 3, the procedure planning results in terms of success rate and mAcc do not show then same trend. It seems like your model perform slightly better or generally the same on COIN rather than on NIV. How can you explain this discrepancy?

---

> ### Author Response · Authors · 2024-11-22
> **Response to reviewer qUxm.**
>
> Dear Reviewer qUxm,
>
> Thank you for your great recognition of our work. We likewise hope that our work can make some modest contributions to the progress of this field. We tried our best to address all the concerns and questions, and update the main paper and appendix(marked blue) in the new version. Please let us know if you have any further concerns or questions to discuss.
>
>
> Best,
>
> Paper 1751 Authors
>
> ___
>
> **W1. Please provide simpler explanations and more visual examples for the interpolation combined with diffusion.**
>
> **A1.** As shown in Figure 2(main paper), our model is based on the U-Net architecture.
>
> The first step is to construct the input matrix $ \hat{x} _ N $ by integrating multiple components: task class labels obtained from our classifier, visual observation features ($V_s$ and $V_g$), and the action sequence $a_{1:T}$. This matrix is then refined through our masked projection during initialization to constrain the action space.
>
> Subsequently, the intermediate latent features $F_j$ generated by the latent space temporal logical interpolation module are incorporated into the diffusion model to progressively refine the input matrix.
>
> The U-Net model, represented by the central gray box in Figure 2(main paper), comprises three downsampling modules, two upsampling modules, and one middle module. Each of these modules contains two residual temporal blocks, which serve as the integration points for features.
>
> As depicted in Figure 3(b)(main paper), the interpolated features derived from the interpolation module undergo a linear transformation to form keys and values ($LF_j$), ensuring dimensional compatibility with the input matrix. Within the residual temporal block module, the input matrix $\hat{x}_n$ is processed through Convolutional Blocks (CB) and combined with timesteps from TimeMLP (TM), resulting in $\left[CB(\hat{x}_n) + TM(t)\right]$ (where t is randomly sampled), which serves as the query. The query and key-value pairs are then processed via Multi-head Attention mechanisms, facilitating the integration of interpolated features and the input matrix. The resulting matrix is further refined through LayerNorm, Linear layers, and Mish activation, before passing through a Convolutional Block to progress to the subsequent residual temporal block.
>
> We've added more detailed visualizations in Appendix D to illustrate the feature maps of interpolated features.

---

> ### Author Response · Authors · 2024-11-22
> **Response to reviewer qUxm.(2)**
>
> **Q1. Balance of current loss function and other possible kinds of loss function.**
>
> **A2.** While applying weights solely to endpoints initially enhances performance by emphasizing these crucial positions, this approach leads to excessive focus on endpoints at the expense of middle actions. To address this limitation, we introduce a task-adaptive masked proximity loss that implements a more balanced weighting strategy - applying stronger supervision at endpoints while gradually relaxing constraints towards the middle through our weighted gradient loss mechanism. This approach helps achieve better equilibrium between endpoint accuracy and intermediate state supervision. As demonstrated in the comparison table below, our method significantly improves action prediction accuracy across all positions, with particularly notable gains in middle actions (showing a 60.6% improvement for $a_3$ compared to 18.2% for $a_5$). These results validate that our approach successfully balances the model's attention between endpoints and intermediate actions.
>
> Table 1: Comparison of action accuracy at each timestep between endpoint weighting and gradient weighting.
> | Action | Weight on both sides | Gradient Weight | Improvement |
> |--------|---------|---------|------------|
> | $a_1$ | 62.32 | 74.26 | +19.2% |
> | $a_2$ | 26.55 | 42.64 | +60.6% |
> | $a_3$ | 21.45 | 34.44 | +60.6% |
> | $a_4$ | 30.76 | 43.03 | +39.9% |
> | $a_5$ | 61.10 | 72.19 | +18.2% |
>
> We explored several alternative loss functions, including Cross-Entropy (CE) and combinations with ranking loss. Our experiments revealed that MSE loss is most effective for this task. The key advantage of MSE loss lies in its ability to model comprehensive distance relationships between actions throughout the entire action matrix. In contrast, CE loss only computes individual position-wise losses, limiting its capacity to capture complex action dependencies. While we attempted to enhance performance by combining MSE with ranking loss, which focuses on adjacent action relationships, this combination did not yield improvements. We attribute this to potential conflicts between the optimization objectives - MSE's holistic distance modeling versus ranking loss's local relationship focus - which may have counteracted each other's benefits.
>
> Table 2: More experiments for exploring different types of loss functions when T=5.
> | Loss | SR↑ |
> |------|-----|
> | Cross Entropy | 7.77 |
> | MSE | _11.89_ |
> | MSE+Ranking | 8.50 |
> | MSE+GW+M | **15.26** |
>
> More information about the ranking loss function:
>
> For each sequence in a batch, let $a \in \mathbb{R}^{T \times A}$ be the probability distribution of predicted actions by the model (where T is the number of time steps and A is the dimension of the action space). Let $\bar{a} \in \mathbb{N}^T$ be the true action index sequence. The steps to calculate the ranking loss are as follows:
>
> 1. Sort the predicted probabilities of actions for each time step t in descending order to obtain the sorted indices:
>    $$R_t = \text{argsort}(a_t) \quad \text{for } t \in [1,T].$$
>
> 2. Find the position (rank) of the true action in the sorted list:
>    $$\text{rank}_t = \text{position}(\bar{a}_t \text{ in } R_t) \quad \text{for } t \in [1,T].$$
>
> 3. Calculate the rank difference between adjacent time steps:
>
>    $$\text{diff} _ t = \text{rank} _ t - \text{rank} _ {t+1} \quad \text{for } t \in [1,T-1].$$
>
> 4. Define an order mask that only calculates the loss when the true action indices satisfy a specific order relationship:
>    $$\text{mask} _ t = \mathbb{1}[\bar{a} _ t > \bar{a} _ {t+1}] \quad \text{for } t \in [1,T-1].$$
>
> 5. The final ranking loss is:
>    $$L_R = \frac{1}{T-1}\sum _ {t=1}^{T-1} \text{ReLU}(\text{diff} _ t \cdot \text{mask} _ t).$$
>
> Where $\text{ReLU}(x) = \max(0, x)$, and $\mathbb{1}[\cdot]$ is the indicator function, which is 1 when the condition is true and 0 otherwise.
>
> The purpose of this loss function design is to ensure that when in the true action sequence, the index of the previous action is larger than the index of the subsequent action ($\bar{a} _ t > \bar{a} _ {t+1}$), the model should give a higher ranking (i.e., a smaller rank value) to the subsequent action. If this rule is violated, a positive loss value will be produced

---

> ### Author Response · Authors · 2024-11-22
> **Response to reviewer qUxm.(3)**
>
> **Q2. About the classification and procedure planning results on COIN and NIV.**
>
> **A3.** There are two possible explanations for this phenomenon:
>
> - **Classification results**: The dataset size significantly impacts classification performance, particularly in terms of task and action diversity. While COIN contains a larger variety of tasks and actions compared to NIV, this actually makes NIV classification more straightforward due to its less complex task structure and more focused action space.
>
> - **Procedure planning performance**: The effectiveness of procedure planning is primarily determined by the average sequence length of actions within tasks. The model generates intermediate action sequences by leveraging visual features from the start ($V_s$) and goal ($V_g$) states. However, the reliability of visual information gradually diminishes towards the middle of the sequence. Given that NIV contains longer action sequences on average, this degradation of visual information in the middle portions results in decreased prediction accuracy.

---

> ### Author Response · Authors · 2024-11-26
> **Response to reviewer qUxm.(4)**
>
> Dear reviewer qUxm,
>
> Thank you very much for your kind and encouraging feedback. We are grateful for your recognition of our work. In response to your request for further clarification on the more ambiguous aspects, we have carefully revised the manuscript to provide a clearer explanation. We hope that these revisions address your concerns, but please do not hesitate to let us know if any points remain unclear. We deeply appreciate your thoughtful engagement and constructive suggestions.
>
> Best,
>
> Paper 1751 Authors

---

> > ### Comment · Reviewer_qUxm · 2024-11-26
> >
> > Dear authors, thank you very much for your revisions. No further questions.

---

> > > ### Author Response · Authors · 2024-11-27
> > >
> > > Thank Reviewer qUxm for the acknowledgment of our reply. We are happy that you are satisfied with our responses to your questions and concerns!

---

### Official Review · Reviewer_pfah · 2024-11-02

**Soundness:** 3
**Presentation:** 2
**Contribution:** 2
**Rating:** 6
**Confidence:** 5

**Summary:**

This paper proposed a diffusion-based method, i.e. Masked Temporal Interpolation Diffusion (MTID) model for procedure planning. The diffusion model part is based on PDPP which takes the start and goal visual feature, action sequence and task class as input. The author proposed a Latent Space Temporal Logical Interpolation Module to generate the intermediate visual features given the start and goal visual features. Additionally, the authors proposed a masked projection and a task-adaptive masked proximity loss. the masked projection excludes the actions that do not belong to the tasks. The task-adaptive masked proximity loss puts more weight on the actions closer to the start and end actions and puts more penalties on the excluded actions. The experiments were performed on Crosstask, Coin, and NIV datasets, and SOTA results were achieved on most metrics of Crosstask and Coin datasets.

**Strengths:**

- This paper addresses the limitation of the previous diffusion model approach, i.e. the temporal dependencies of actions, by using generated intermediate visual supervision.
- The Latent Space Temporal Logical Interpolation Module seems to be helpful as the intermediate visual supervision.
- The task-adaptive masked proximity loss seems to contribute to the increase of performance.

**Weaknesses:**

- The Latent Space Temporal Logical Interpolation module does not have anything to do with 'logic'. It only has temporal dependencies between actions. Linear interpolation between start and goal visual observations with weights is not logic.

- The experiment setting needs to be specified clearly in the main manuscript. The results on Coin and NIV datasets are misleading. This paper follows the setting of PDPP, however the results of baseline models (KEPP, SCHEMA) follow a different experiment setting. This needs to be clarified. Furthermore, the NIV dataset misses the results of PDPP, the proposed method does not outperform PDPP on all metrics for both time horizons.

- Although the proposed ideas are interesting, it is not clear how the Latent Space Temporal Logical Interpolation module and the task-adaptive masked proximity loss contribute to the improvement on performance as they are built on top of PDPP. It will be beneficial to see the ablation studies of different combinations of model components on all datasets, i.e. diffusion + Latent Space Temporal Logical Interpolation, diffusion + masked project + task-adaptive masked proximity loss.

- The presentation of the work has ambiguities and needs to be clarified. For details see comments below.

**Questions:**

In the paragraph starting from line 81, the authors need to clarify which parts in MTID are from PDPP (e.g. using predicted task class in the input matrix for the diffusion model, mask intermediate visual features, etc.)

- In equation 1, why introduce M? v1:M should just be v2:T-1, i.e. the number of intermediate steps.

- The notations for actions are inconsistent. Sometimes \hat{a_t} is the ground truth action, sometimes \bar{a_t} is (e.g. Eq 10), and sometimes a_t is (e.g. Figure 2). Please make them consistent throughout the full text.

- The caption of Figure 2 needs to be elaborated. And the part to exclude actions that do not belong to a class is not shown. The fonts are too small to read.

- The caption of Figure 3 needs to be elaborated. The fonts are too small to read.

- Line 316, PDPP weights both start and end actions.

- Put What does the dagger symbol mean in the caption of Table 1 too.

- Line 359, Crosstask has 105 action classes.

- Merge Table 13 in the appendix to Table 3 in the main manuscript to show results on all metrics on Coin and NIV.

---

> ### Author Response · Authors · 2024-11-22
> **Response to reviewer pfah.**
>
> Dear Reviewer pfah,
>
> We sincerely thank the reviewer for such careful attention in our paper, and we express our genuine respect for the reviewer's thorough and responsible approach. The latest version of the paper(marked orange) has been submitted. Please let us know if there are any further issues.
>
> Best,
>
> Paper 1751 Authors
>
> ___
>
> **W1. The interpolation module is not logical.**
>
> **A1.** Specifically, the "**Latent Space Temporal Logical Interpolation**" module is composed of "Observation Encoder", "Latent Space Interpolator" and "Transformer Encoder Blocks" in Figure 3(a).
>
> 1. Observation Encoder: This part is used to extract the features into latent space, to better capture abstract representations and learn meaningful feature embeddings that are more suitable for our tasks, corresponding to "**Latent Space**".
>
> 2. Latent Space Interpolator: The function of this part is to create new visual features based on $V_s$ and $V_g$, which not only add more diverse temporal characteristics but also preserve the original information, similar to the interpolation of video frames[1]. According to the temporal logic of actions (TLA)[2], which is a logic for specifying and reasoning about concurrent systems, temporal relationships can be considered as a type of logical relationships. This part corresponds to the preliminary modeling of **Temporal** logical relationships using **Interpolation**.
>
> 3. Transformer Encoder Blocks: According to Hahn et al.[3], our transformer encoder blocks learn the underlying semantics of these relationships and enhance the logical modeling by incorporating both temporal and causal dependencies into the features, corresponding to **Logical**.
>
> In fact, actions have not only temporal relationships but also causal and other relationships. Therefore, we use "logical" to emphasize that our model considers both temporal and causal dependencies between actions.
>
> [1] Danier, D., Zhang, F., & Bull, D. (2024). LDMVFI: Video frame interpolation with latent diffusion models. In Proceedings of the AAAI Conference on Artificial Intelligence (Vol. 38, No. 2).
>
> [2] Lamport, Leslie. "The temporal logic of actions." ACM Transactions on Programming Languages and Systems (TOPLAS) 16.3 (1994): 872-923.
>
> [3] Hahn, C., Schmitt, F., Kreber, J. U., Rabe, M. N., & Finkbeiner, B. (2020). Transformers generalize to the semantics of logics. arXiv preprint arXiv:2003.04218.
>
> ___
>
> **W2. The experiment settings and results need clarification.**
>
> **A2.** In the latest version of the paper, we have clarified the experimental settings and results.
>
> For CrossTask, the results in Table 1 (main paper at line 324) are correct under the experiment settings of PDPP, demonstrating the effectiveness of our approach on this dataset.
>
> For COIN and NIV datasets, we have unified the experimental settings across all methods using KEPP's setting and presented our results in Table 3 (main paper at line 384). The evaluation results demonstrate that our approach consistently outperforms the best-performing methods on both datasets, highlighting our model's robust performance across datasets of different sizes.
>
> We also present the results on COIN and NIV under the settings of PDPP in Appendix D, section "Comparison with PDPP" at line 1044. For NIV, our performance is slightly lower, which can be attributed to two main factors:
>
> 1. Dataset size: NIV is significantly smaller than CrossTask and COIN, which leads to the model excessively learning detailed patterns from the training data, resulting in reduced generalization ability.
>
> 2. Experimental setting differences: PDPP defines states as the window between start and end times, while KEPP uses a 2-second window around start/end times. This difference allows PDPP to access more step information, especially for short-term actions, potentially weakening the impact of our interpolation feature supplementation.
>
> Despite these challenges with NIV under PDPP settings, it's important to note that our model demonstrates strong capabilities on larger CrossTask and COIN, showcasing its effectiveness in temporal logic and memory utilization.

---

> ### Author Response · Authors · 2024-11-22
> **Response to reviewer pfah.(2)**
>
> **W3. The paper needs ablation studies to show how each proposed component contributes to the performance.**
>
> **A3.** We have added the relevant experiments in Appendix C under the section "Ablation for Our Different Methods" at line 971. The experimental results in Table 1 demonstrate that our proposed components significantly improve the model's performance. Other experiments on COIN and NIV are also provided in Appendix C at line 972.
>
> Table 1: Ablation study with our proposed components on CrossTask.
> | ID  | M | K | L | SR↑   | mAcc↑  | mIoU↑  |
> | --- |---|---|---|-------|--------|--------|
> | 1   |   |   |   | 37.20 | 64.67  | 66.57  |
> | 2   | ✓ |   |   | 39.03 | 66.49  | 68.26  |
> | 3   |   | ✓ |   | 38.88 | 66.36  | 68.35  |
> | 4   |   |   | ✓ | 38.57 | 66.02  | 68.17  |
> | 5   | ✓ | ✓ |   | 39.64 | _66.74_ | 68.77  |
> | 6   | ✓ |   | ✓ | _39.71_ | 66.65  | _68.83_ |
> | 7   |   | ✓ | ✓ | 39.17 | 66.49  | 68.38  |
> | 8   | ✓ | ✓ | ✓ | **40.45** | **67.19** | **69.17** |
>
> Note: M: Our latent space temporal logical interpolation module, K: mask projection, L: task-adaptive masked proximity loss. The results of ID 1 are from PDPP.
>
>
>
> ___
>
> **Q1. Clarify which components in MTID are inherited from PDPP.**
>
> **A4.** Our method builds upon the base PDPP model, inheriting several components while addressing its limitations. Here are the inherited components and our improvements:
>
> 1. **Input Matrix**:
>    - Inherited: We maintain the same input matrix design.
>    - Drawback: This design leads to the model predicting actions that are less likely to fall within the current task label's action space.
>    - Improvement: We introduce masked projection during initialization to limit the action space and reduce the probability of predicted actions falling outside the action space.
>
> 2. **U-Net Architecture**:
>    - Inherited: We adopt the base U-Net model from PDPP.
>    - Drawback: This architecture is too simple to capture the temporal logic of actions and lacks intermediate visual features for supervision.
>    - Improvement: We extend it by incorporating a latent space temporal logical interpolation module and a cross-attention module, enhancing temporal reasoning and feature integration.
>
> 3. **Loss Function**:
>    - Inherited: We retain PDPP's base MSE loss while removing the weights on both sides.
>    - Drawback: This loss is more likely to cause predicted actions to fall outside the current task's action space. Additionally, the real information provided by visual features gradually weakens - the farther from the endpoints, the worse the effect becomes, as shown in Figure 6d. Weighting only both ends imposes excessive attention on endpoints, causing over-reliance while neglecting the middle portions.
>    - Improvement: We introduce additional constraints, including gradient weighting and masking techniques, to enhance the model's performance and training stability.
>
> 4. **Diffusion Process**:
>    - Inherited: We retain the denoising diffusion process.
>    - Drawback: This diffusion process is time-consuming.
>    - Improvement: We replace DDPM with DDIM, which accelerates both training and inference while maintaining high-quality predictions.
>
> 5. **Task Classification**:
>    - Inherited: We maintain a task classifier component.
>    - Drawback: MLP classifier is too simple to model complex spatial-temporal relationships between actions for long instructional videos to obtain accurate task labels.
>    - Improvement: We enhance it by integrating a transformer module, improving the model's ability to understand and classify complex tasks through transformer's long-term temporal relationship modeling capabilities.
>
> These improvements address PDPP's limitations in temporal reasoning, feature integration, and computational efficiency, resulting in a more powerful and versatile model for procedural planning in instructional videos.

---

> ### Author Response · Authors · 2024-11-22
> **Response to reviewer pfah.(3)**
>
> **Q2. In equation 1, why introduce M?**
>
> **A5.** Our MTID diffusion model takes as input a matrix containing action sequences with T timesteps and is based on U-Net, which contains **M residual temporal blocks** in the downsampling, upsampling, and middle layers for directly diffusing and generating T intermediate target actions. To ensure that each intermediate layer contains valid auxiliary information, our Latent Space Temporal Logical Interpolation Module needs to generate M intermediate auxiliary features. Subsequently, we apply cross-attention in residual temporal blocks across the M interpolated features and the entire input matrix rather than individual timesteps, enabling better temporal integration.
>
> We also conducted experiments to demonstrate the effect of M. Our results showed that using interpolated features only for T steps led to suboptimal performance. This also supports our decision to use interpolated features across all M modules.
>
> Table 2: Ablation study on M on CrossTask when T=3.
> | Method | SR↑ | mAcc↑ | mIoU↑ |
> |--------|-----|--------|--------|
> | T | 38.64 | 66.13 | 68.05 |
> | M | **40.45** | **67.19** | **69.17** |
>
> About more information, we have added the relevant experiments in Appendix D, section "More Explanation of M" at line 990.
>
> ____
>
> **Q3. Some mistakes in the paper.**
>
> **A6.** We have corrected the mistakes in the latest version of the paper.

---

> ### Author Response · Authors · 2024-11-26
> **Response to reviewer pfah.(4)**
>
> Dear reviewer pfah,
>
> Thank you for your detailed and constructive feedback. We hope that our revisions and responses have addressed the points you raised, particularly regarding the minor errors and additional experimental work. We have carefully incorporated the suggested corrections and are happy to provide any further clarifications if necessary. Please feel free to share any additional comments or suggestions.
> We greatly appreciate your thorough review and continued support.
>
> Best,
>
> Paper 1751 Authors

---

> > ### Comment · Reviewer_pfah · 2024-12-01
> >
> > Thanks for the revision, most of my comments are addressed. However, I am not convinced by the 'logic' of the Latent Space Temporal Logical Interpolation module, Yes, I agree actions have temporal and causal relations. But the module is logical, it means that one can deploy programs (e.g. consists of binary operations like AND, OR and so on) to infer actions. Clearly Latent Space Temporal Logical Interpolation module can't do this. I wouldn't link anything to logic, it only creates confusion. Moreover, causal relation is also not the Latent Space Temporal Logical Interpolation module learns. It requires a model to learn invariant representations which satisfy identifiability, such as causal representation learning methods. I will not link the Temporal Logical Interpolation module to causality either.
> >
> > Overall, I appreciate the additional ablation studies. I still have the concern that the proposed method MTID does not consistently outperform SOTA on all metrics on all datasets (SR when T=6 on crosstalk, SR when T=3 on coin). But I will increase my score.

---

> > > ### Author Response · Authors · 2024-12-03
> > > **Response to reviewer pfah.(5)**
> > >
> > > We sincerely appreciate the reviewer's recognition of our work and the increased score.
> > >
> > > Regarding the Latent Space Temporal Interpolation module, we have decided to remove the 'logical' word from the paper to avoid confusion. It also should be noted that the module is not designed for concrete but implicit logical and casual inference.
> > >
> > > For the comparison with state-of-the-art methods, we would like to address several points:
> > >
> > > - First, regarding the comparison with SCHEMA[1], their use of Large Language Models (LLMs) introduces external knowledge not present in the original dataset, making it an unfair comparison.
> > >
> > > - Second, for COIN with T=3, our analysis in Table 9 (line 843) demonstrates that increasing model size alone can improve performance. This finding suggests that for larger datasets, it is essential to develop a memory mechanism to store relevant information, rather than focusing solely on temporal relationships. We will explore this direction in our future research.
> > >
> > > - Finally, our analysis in Table 15 (line 1012) shows that for T=6 compared to T=5, there is a larger gap between real and interpolated features. This suggests our current interpolation strategy may be insufficient for longer sequences, as features lack diversity to capture actions fully. We will explore improving interpolated feature quality in future work.
> > >
> > > [1] Niu, Y., Guo, W., Chen, L., Lin, X., & Chang, S. F. (2024). SCHEMA: State CHangEs MAtter for Procedure Planning in Instructional Videos. arXiv preprint arXiv:2403.01599.

---

### Meta-Review · Area_Chair_zhqy · 2024-12-24

**Metareview:**

The paper presents the Masked Temporal Interpolation Diffusion (MTID) model, a novel approach for procedure planning in instructional videos. By introducing a latent space temporal interpolation mechanism within a diffusion framework, MTID generates intermediate visual features that enhance temporal reasoning and state transitions. The paper demonstrates strong empirical performance, achieving state-of-the-art results on major datasets like CrossTask, COIN, and NIV, with consistent improvements in most key metrics. MTID’s innovations include task-adaptive masked proximity loss, a transformer-based task classifier, and the integration of masked projections, all contributing to improved modeling of complex temporal dependencies.

The reviewers generally recognized the paper’s significant contributions. They praised the innovative use of interpolation to bridge the gaps between observed and unobserved states, the novel loss design that balances endpoint supervision with intermediate state accuracy, and the model’s robust performance across datasets. The proposed approach effectively addresses limitations of previous models, such as PDPP, by incorporating visual-level supervision and enhancing the temporal reasoning capabilities with cross-attention mechanisms. The ablation studies further highlighted the impact of the individual components, demonstrating that each innovation contributed meaningfully to the overall performance.

Given the paper’s robust experimental results, innovative methodological contributions, and the extensive efforts to address reviewer feedback, the AC recommends acceptance. The work advances the state of the art in procedure planning for instructional videos. The remaining concerns are minor and do not detract from the paper’s significant strengths and practical impact.

**Additional Comments On Reviewer Discussion:**

Please refer to the meta-review

---

### Decision · Program_Chairs · 2025-01-22

Accept (Poster)